# Remapping of Greenland ice sheet surface mass balance anomalies for large ensemble sea-level change projections

Heiko Goelzer[1,2], Brice P. Y. Noel[1], Tamsin L. Edwards[3], Xavier Fettweis[4], Jonathan M. Gregory[5,6], William H. Lipscomb[7], Roderik S. W. van de Wal[1,8], Michiel R. van den Broeke[1]

(1) Institute for Marine and Atmospheric research Utrecht, Utrecht University, Utrecht, the Netherlands
(2) Laboratoire de Glaciologie, Université Libre de Bruxelles, Brussels, Belgium
(3) Department of Geography, King's College, London, UK
(4) Laboratory of Climatology, Department of Geography, University of Liège, Liège, Belgium
(5) National Center for Atmospheric Science, University of Reading, Reading, UK
(6) Met Office, Hadley Centre, Exeter, UK.
(7) Climate and Global Dynamics Laboratory, National Center for Atmospheric Research, Boulder, CO, USA
(8) Geosciences, Physical Geography, Utrecht University, Utrecht, the Netherlands

Correspondence to: Heiko Goelzer (h.goelzer@uu.nl)

## Abstract

Future sea-level change projections with process-based standalone ice sheet models are typically driven with surface mass balance (SMB) forcing derived from climate models. In this work we address the problems arising from a mismatch of the modelled ice sheet geometry with the one used by the climate model. We present a method to apply SMB forcing from climate models to a wide range of Greenland ice sheet models with varying and temporally evolving geometries. In order to achieve that, we translate a given SMB anomaly field as a function of absolute location, to a function of surface elevation for 25 regional drainage basins, which can then be applied to different modelled ice sheet geometries. The key feature of the approach is the non-locality of this remapping process. The method reproduces the original forcing data closely when remapped to the original geometry. When remapped to different modelled geometries it produces a physically meaningful forcing with smooth and continuous SMB anomalies across basin divides. The method considerably reduces non-physical biases that would arise by applying the SMB anomaly derived for the climate model geometry directly to a large range of modelled ice sheet model geometries.

# 1        Introduction

Process-based ice sheet model projections are an important tool to estimate future sea-level change in the context of the Intergovernmental Panel on Climate Change assessment cycle (IPCC, 2013). For the first time, in the upcoming IPCC assessment report (AR6), ice sheet model projections are formally embedded in the Coupled Model Intercomparison Project (CMIP, Eyring et al., 2016) in the form of the CMIP-endorsed Ice Sheet Model Intercomparison Project ISMIP6 (Nowicki et al., 2016; 2020). ISMIP6 aims at providing estimates of the future sea-level contribution from the Greenland and Antarctic ice sheets based on standalone ice sheet model (ISM) simulations, forced by output from CMIP atmosphere-ocean global climate models (GCMs) and fully-coupled ISM-GCMs. This paper focuses on standalone simulations of the Greenland ice sheet (GrIS).

The first ISMIP6 activities focused mainly on the problem of ice sheet model initialisation (Goelzer et al., 2018a; Seroussi et al., 2019), but also identified issues that may be encountered when a large range of ice sheet models is forced with climate model output. The most important forcing derived from climate models in the context of future sea-level change projections for the GrIS is the surface mass balance (SMB) describing the rate at which mass is added or removed at the ice sheet surface. For the ISMIP6 projections it was decided to apply the SMB forcing as an anomaly, i.e. as the change in SMB relative to a given reference period. This approach has the important advantage that it allows for participating ice sheet modellers to use their own SMB product during initialisation and simply add provided SMB anomalies in a projection experiment.

However, problems were identified when a given surface mass balance anomaly (aSMB) was applied to the wide range of Greenland ice sheet models used in the community (Goelzer et al., 2018a). The key issue is a mismatch between modelled initial and observed ice sheet geometries, the latter of which underlies the SMB field. These differences are related to uncertainties in forcing, physical parameters, and the underlying ice sheet model physics. For instance, a geometrical mismatch generally means that the modelled ablation zone and the prescribed anomalous ablation are not co-located, leading to an incorrect mass balance forcing.

With the original intention to apply identical forcing to all participating models, a forcing data set was prepared for initMIP-Greenland (Goelzer et al., 2018a) that consisted of an SMB anomaly based on the present-day observed geometry. The SMB anomaly was extended outside the observed ice sheet mask following a simple parameterization to accommodate larger than observed ice sheet model extents. In practice, however, ice sheet models with larger-than-observed initial areas exhibit larger melting under such forcing, simply because their ablation areas are extended outwards.

To address this problem, we present here a method to remap the SMB anomaly as a function of surface elevation, and thereby produce physically consistent forcing for different ice sheet model geometries. The proposed method was developed for future sea-level change projections made with a large ensemble of ice sheet models (with possibly widely different initial geometries) forced by output of different climate models and scenarios. However, other applications can be envisioned, for example any

other case where the climate model forcing is generated for an ice sheet geometry differing from that of the ice sheet model itself. Asynchronously-coupled climate-ice sheet simulations and experiments with accelerated climatic boundary conditions may also be improved with the presented method.

In the following we describe our approach and method (Sec 2), the resulting forcing (Sec 3), and time dependent applications (Sec 4), and finally discuss the results (Sec 5).

## 2  Approach and method

Our approach aims to generate a SMB forcing (at a yearly time scale) applicable to an ensemble of Greenland ice sheet models that exhibit a wide range of initial present-day ice sheet geometries. The forcing is based on an existing aSMB product that is generated at a fixed present-day surface elevation. This aSMB product will typically be the output of a regional climate model, but could come from any SMB model or GCM. While the forcing will have to be adapted for the individual model geometries, it should remain as close as possible to the original product when applied to the observed present-day geometry.

The proposed method is based on the strong elevation dependence of SMB and aSMB and is illustrated for a schematic flowline of a land-terminating ice sheet margin (Figure 1). For a larger ice sheet geometry (red, dashed), the horizontal equilibrium line position lies farther from the ice divide than for a smaller ice sheet (black). It is this effect that we are trying to capture with our method: a different ice sheet geometry requires a different forcing to honour physical consistency. Remapping the SMB anomaly as a function of surface elevation, as we propose, allows for a "stretching" of the SMB product to match the larger ice sheet extent, while maintaining its overall shape.

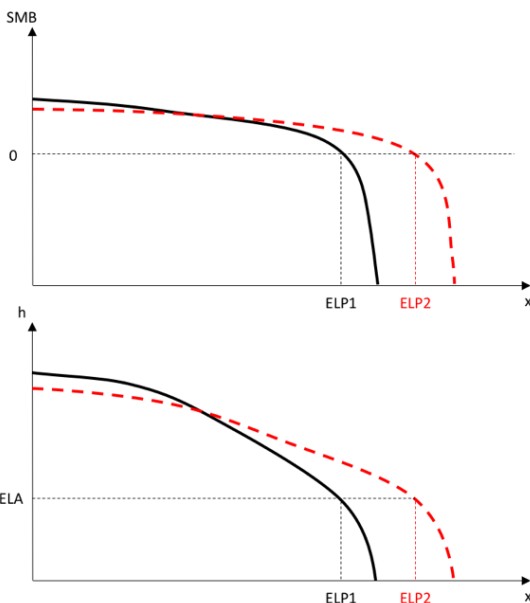

**Figure 1 Schematic cross section for two different ice sheet geometries (bottom) and associated surface mass balance (top). The two geometries share the same equilibrium line altitude (ELA), but exhibit different horizontal equilibrium line positions (ELP1, ELP2).**

For initMIP-Greenland, the SMB anomaly was parameterised as a fixed function of observed surface elevation and latitude

sampled across the entire ice sheet (Goelzer et al., 2018a), which was subsequently used to define a forcing product everywhere on the grid. In principle, we could use the same global approach to generate SMB forcing for a range of different initial ice sheet geometries. However, regional differences in the height-aSMB relationship can be large and justify a spatially better resolved approach.

To capture regional differences, we therefore apply the remapping separately for a set of drainage basins (Shepherd et al.,

2012; Zwally et al., 2012; Mouginot et al. 2019). In practice, the following steps are executed to (1) derive and (2) apply the height-aSMB relationship to different geometries.

(1) Defining an elevation-aSMB lookup table:

- Divide the ice sheet into drainage basins
- For each individual drainage basin do:

- For each elevation band with central height $h_c$ and range R of heights do:
    - find aSMB values for all heights in R
    - calculate the median aSMB of these
    - Save result to lookup table aSMB=f($h_c$)

(2) Remap aSMB to a new geometry:

- Use the drainage basins separation in (1)
- For each individual drainage basin do:

- For each ISM grid point do:
  - interpolate aSMB linearly as a function of height using a combination of lookup tables (1) for this and neighbouring basins (see Sec 2.2)

## 2.1  Defining an elevation-aSMB lookup table

The first step (defining an elevation-aSMB lookup table) is independent of the ice sheet model characteristics and relies only on the initial aSMB product, the reference field's elevation, and a meaningful basin selection. Ideally, the basin division should separate regions with largely different SMB characteristics, e.g. wet and dry regions. At the same time, our method requires that each basin contains a wide elevation range so that the lookup tables can be completely filled. For this study we created 25 basins by combining several smaller basins from a recent drainage delineation (Mouginot et al. 2019). The basins may consist

only of single outlet glaciers or even flowlines, as long as they cover a sufficiently large elevation range. The basin delineation is extended outside the observed ice sheet mask to accommodate different (i.e. larger) ice sheet geometries than observed (Figure 2). This was done once manually using observed topography of ice-free regions and bathymetry as guidance. In order to test the robustness of the method to the number of basins, we have constructed an alternative basin set that can be subdivided semi-automatically, albeit not following observed drainage divides (Figure S1, supplementary material).

While the method can be applied to any aSMB product, here we use model output from the regional climate model MAR (Fettweis et al., 2013) forced by MIROC5 (Watanabe et al. 2010), as it has been run for the RCP8.5 scenario and was chosen for ISMIP6. We use output of MAR version 3.9 run at a horizontal resolution of 15 km that has been downscaled to 1 km (Delhasse et al., 2019) and subsequently interpolated to 5 km resolution for our analysis. If needed e.g. for a coarser resolution climate model output, the aSMB could be interpolated to a high enough target resolution to guarantee that sufficient samples

are present in each basin and elevation band. We demonstrate the method here with aSMB at the end of the century relative to the 1960-1989 reference period, calculated as the time mean change:

$$aSMB = \overline{SMB}^{2091-2100} - \overline{SMB}^{1960-1989}. \tag{1}$$

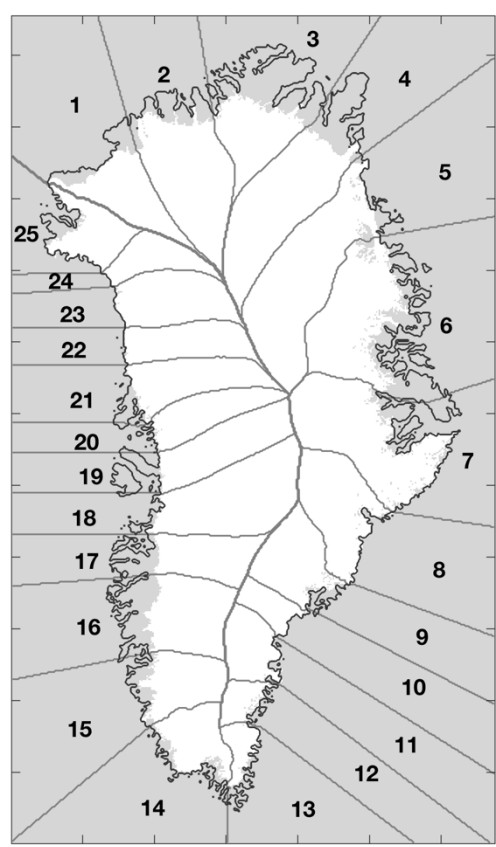

**Figure 2 Basin separation. The basin delineation is based on Mouginot et al. (2019), combined into a set of 25 regional basins and extended to the grid margin.**

5    For each drainage basin we define an elevation-aSMB lookup table based on the MAR SMB data in that basin. We define elevation bands with centre $h_c$ and range R, find all grid points with matching elevation, and register the associated aSMB values. We calculate the median aSMB value of all available points for each elevation band (Figure 3), resulting in a lookup table aSMB=f(hc). The median is chosen rather than the mean for its robustness to outliers. The step size dh=100 m between subsequent elevations $h_c$ and the value for the range of R=100 m was chosen after some initial testing, but was not formally

10   optimised. The main factors influencing this parameter choice are spatial variability and smoothness of the original aSMB product, which also depends on the original resolution of the SMB model (in this case: 15 km). Given the relatively smooth aSMB field, the chosen parameters were judged sufficient to describe the variation in the elevation-aSMB relationships for each basin (Figure 3). Other interval sizes may be more appropriate for other climate forcing products.

For all table entries at 0 m elevation, we have copied the more robust table entry at 100 m, rather than using the 0-50 m height

15   interval with sparser data. For basins with missing values for high elevations, we repeated the highest-elevation aSMB value until 3500 m (circles in Figure 3).

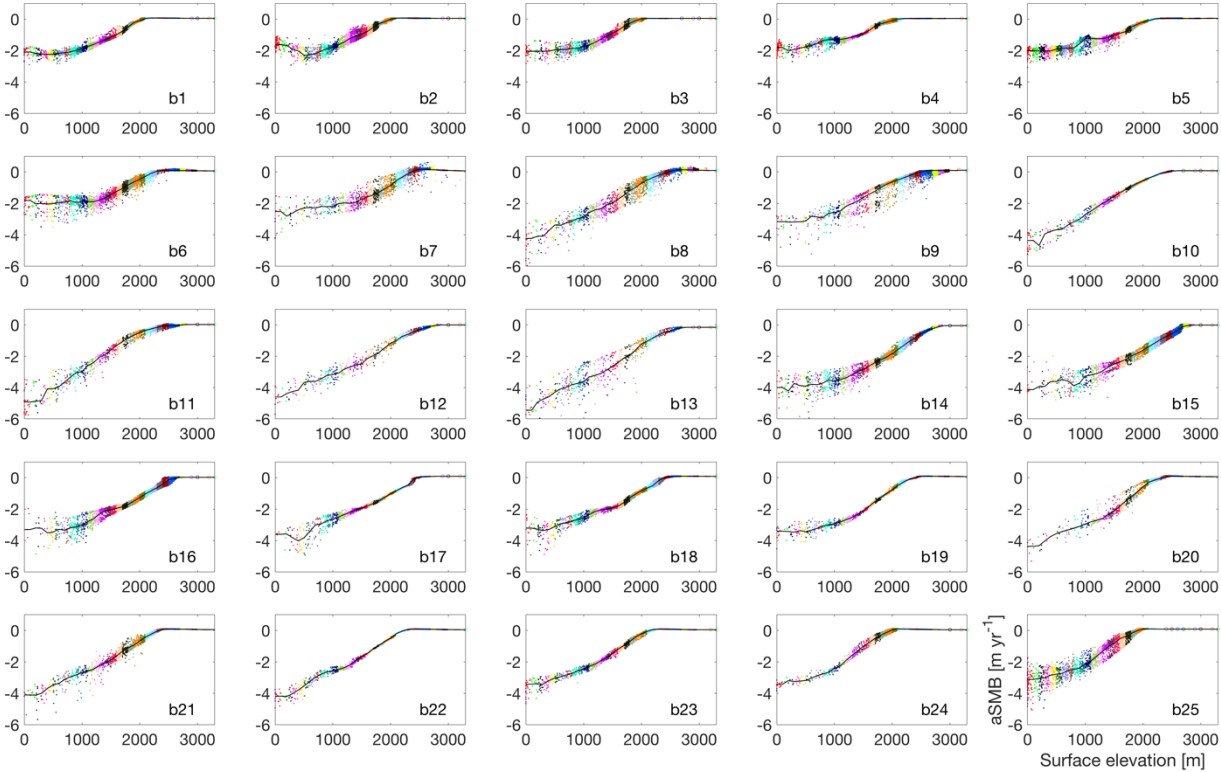

**Figure 3 SMB anomaly (m ice equivalent per year) from the RCM MAR (scatter) and with the elevation interval medians (used for the mapping) shown with a black line. Different colours indicate the elevation ranges considered for the elevation-aSMB lookup table. The subfigure labels indicate the basin identifiers as defined in Figure 2**

## 2.2 Remap aSMB to a new geometry

For the reconstruction of SMB on an ice sheet model geometry, we define the aSMB for each grid point using a combination of lookup tables from the local and neighbouring basins. We weight the aSMB values of the surrounding neighbour basins by proximity, which results in a gradual decrease of influence of the next neighbouring basin away from the divides (Figure 4). The aSMB for each point in a specific basin $b_0$ is calculated as

$$aSMB_{b0}(x,y) = aSMB_{b0}(h) * w_0(x,y) + aSMB_{b1}(h) * w_1(x,y) + \dots aSMB_{bn}(h) * w_n(x,y), \tag{2}$$

where $aSMB_{bi}(h)$ is the aSMB value found by interpolating the lookup table for basin $b_i$ at the elevation $h(x,y)$.

The weights of the gradients in the current basin $b_0$, are calculated as

$$w_0 = 1 - \frac{p_1 + p_2 + \dots + p_n}{p_0 + p_1 + p_2 + \dots + p_n}, \tag{3}$$

which is the residual of the sum of the weights for neighbouring basins $b_1$ through $b_n$ defined as

$$w_1 = \frac{p_1}{p_0 + p_1 + p_2 + \cdots + p_n} \tag{4}$$

$$\cdots$$

$$w_n = \frac{p_n}{p_0 + p_1 + p_2 + \cdots + p_n}.$$

Here $p_0 = 1$ and $p_1, p_2, \ldots p_n$ are proximities of a given point to the neighbouring basins $b_1 - b_n$ which are limited to the interval [0, 1]:

$$p_i = 1 - \min\left(\frac{ds_i}{ds_{norm}}, 1\right), \tag{5}$$

where $ds_i$ is the distance from a given point in $b_0$ to the nearest point in neighbouring basin $b_i$, which is normalized by a prescribed distance $ds_{norm} = 50\ km$. This value of $ds_{norm}$ was chosen to minimize the mismatch between original and reconstructed aSMB (other tested values were 75, 100 and 125 km), though variations in $ds_{norm}$ have limited influence on the results. As an example, near divides with only one neighbouring basin in proximity, the local weighting factor $w_0$ increases from 0.5 at the divide to 1.0 at the centre of the basin (Figure 4).

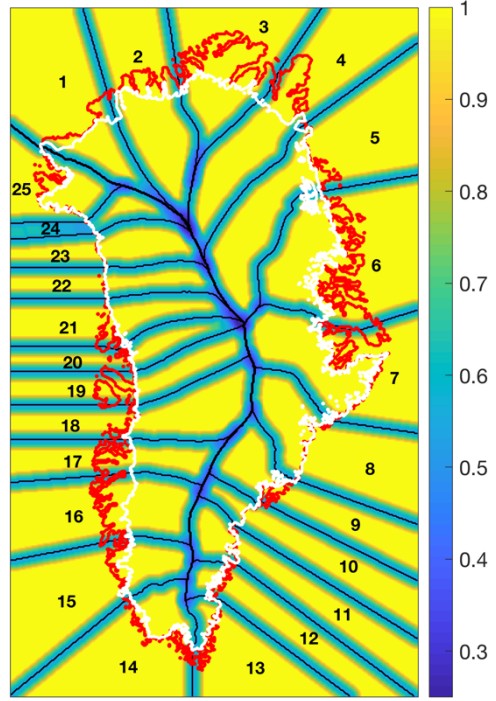

**Figure 4 Weighting factor of the local basin for remapping. The local weighting factor increases from the basin divides (black lines) to 1.0 in the centre over a specified distance (here 50 km), while the factor for the neighbouring basin decreases proportionally (not shown). The white contour outlines the ice sheet margin and the red line the Greenland coast.**

# 3        Results

Figure 5 shows results for aSMB at the end of the MAR RCP8.5 simulation (Eq. 1). The original MAR aSMB (Fig. 5a) has been used to remap aSMB at the same surface elevation (Fig. 5b).

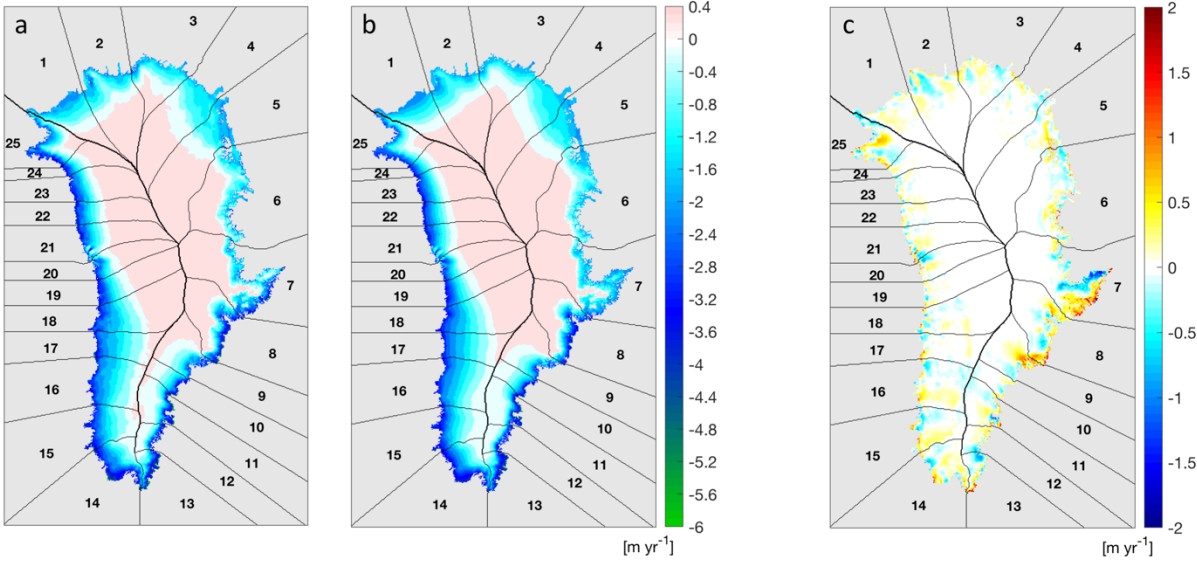

**Figure 5 SMB anomaly from the RCM MAR for the observed geometry (a), remapped to the same observed geometry (b) and differences (b)-(a) in (c).**

The reconstructed aSMB is very similar to the original, reproducing the overall pattern. Some smaller-scale features are lost, however, by averaging laterally across the basin and over elevation bands. The difference map (Fig. 5c) reveals some along-flow features at the margins (e.g. in basins 2, 3, 9, 15, 16  and 17), suggesting that the local median value is not a good representation and that refinement of those basins could further improve the remapping. The absolute error in spatially integrated aSMB per region in this case is on average 2.3% with extremes of 4%, 6% and 16% in basins 5, 8 and 9, respectively (Figure 6). These three basins all exhibit detailed and varied topography at the margins, which may contribute to the errors. The largest signed errors are found in basin 7 with compensating biases of opposite sign. We consider these errors acceptable given typical uncertainties in climate model forcing (e.g. van den Broeke et al., 2017) and our specific interest in large scale, ice-sheet-wide results to be used in ISMIP6. Specifically, the aSMB error integrated over all basins is 18 km$^3$ yr$^{-1}$ (Figure 6) compared to an ensemble range (650 km$^3$ yr$^{-1}$) and ensemble standard deviation (240 km$^3$ yr$^{-1}$) for the 6 CMIP5 models used in ISMIP6 (Goelzer et al., 2020). The robustness of the method to changes in the number of basins has been evaluated with a schematic basin set that can be subdivided semi-automatically (Supplementary material). Within the range of tested basin numbers (20-100) the remapping error is the lowest for the largest number of basins (100), but varies non-steadily and by only up to 15 % across the tested range (Figure S2).

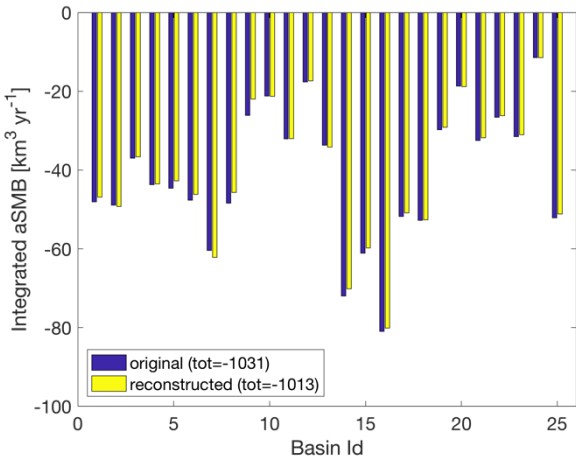

**Figure 6 Integrated aSMB per basin from original MAR model output (blue) and for reconstruction on the same geometry (yellow). Greenland-wide total values are given in the legend.**

The remapped aSMB for an example modelled geometry with large differences relative to the observed is shown in Figure 7c for one member of the initMIP ensemble (VUB_GISM). The remapped aSMB shows a pattern similar to the original (Figure 7a) with smooth and continuous aSMB across basin divides. Where the ice sheet extends well beyond the observed ice mask (grey contour lines) the aSMB is naturally extended following the modelled surface elevation, as is best visible in sector 3. Results from a standard method of extending the SMB outside the observed ice sheet mask at the observed surface elevation

(Franco et al., 2012) are shown in Figure 7b for the footprint of the modelled ice sheet. This method uses the 4 closest, distance-weighted SMB values inside the MAR ice mask, and applies a correction based on the elevation difference between the interpolated elevation of the 4 SMB pixels and the local elevation by using the local vertical SMB gradient computed in this area. Due to low elevation of the tundra surrounding the ice sheet, the extension provides generally low aSMB for regions outside the observed ice sheet mask, which is illustrated in Figure 7d, showing the difference between the original (Figure 7a)

and extended (Figure 7b) aSMB. By definition, the original and extended aSMB are identical over the common ice mask, but positive differences can be seen in regions where the modelled ice sheet is smaller (e.g. basin 16, Figure 7d). The remapping method notably prevents the occurrence of large-amplitude negative aSMB outside of the observed ice sheet mask, illustrated by the difference between the two approaches (Figure 7e).

We quantify the differences between the three aSMB products again by integrating them over the drainage basins (Figure 8a).

The largest differences between the original and extended aSMB are found in basins where the modelled ice sheet extends far beyond the observed ice sheet mask (basins 3, 4 ,6 and 7), or where the aSMB has large negative amplitude (basin 12, 14 and 15). In all these cases, the remapping reduces the bias (in most cases considerably), which is visualised by showing basin integrals of differences between original and extended (blue) and between remapped and extended aSMB (yellow) in Figure

8b. In most cases, biases in the extended aSMB (blue) are reduced by the remapping, illustrated by bars of the same sign (yellow).

The biases are reduced but are not expected/supposed to be entirely removed by the remapping, because a physically larger ice sheet should have a larger accumulation and/or ablation areas. This also illustrates why the method is not designed to conserve mass when remapping to a different geometry: it demands a different SMB forcing. The improvement of the aSMB forcing by the remapping is mainly found in regions where the modelled ice sheet extends beyond the observed mask and where the remapped aSMB is predominantly higher than the extended aSMB (Figure 7e). Differences between original and remapped aSMB in the interior of the ice sheet (Figure 7e) indicate averaging in the remapping process as discussed before, but more importantly are due to differences in the modelled surface elevation compared to the observed. This illustrates a feature of the remapping method that can be interpreted both as an asset or as a shortcoming, namely that biases in surface elevation (Figure 7f) are propagated to the aSMB forcing.

For ice sheet models with initial states close to observations, the reconstructed aSMB looks very similar to the original, while for models with largely different geometry, the overall structure of decreasing aSMB towards lower elevation is well captured. A similar comparison as in Figure 7c and Figure 8a, for three other modelled geometries from the initMIP-Greenland ensemble is given in the supplement (Figure S3 and Figure S4).

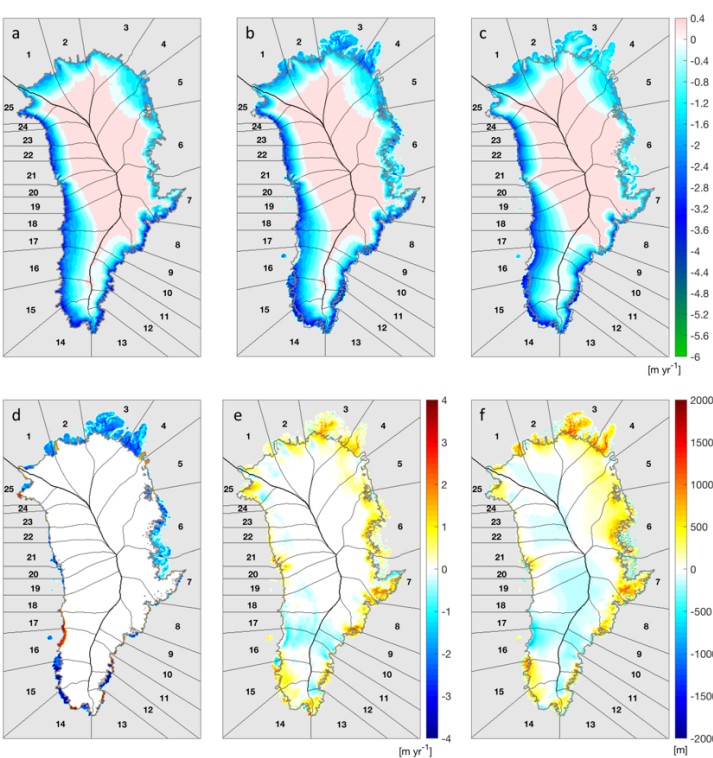

**Figure 7 (a) SMB anomaly from the RCM MAR (same as Figure 5a), (b) extended to the VUB_GISM initial geometry using the method of Franco at al. (2012), (c) remapped with weighting between neighbouring basins for the same geometry, (d) difference b-a, (e) difference (c)-(b) and (f) model bias in surface elevation. The grey lines mark the observed ice sheet margin.**

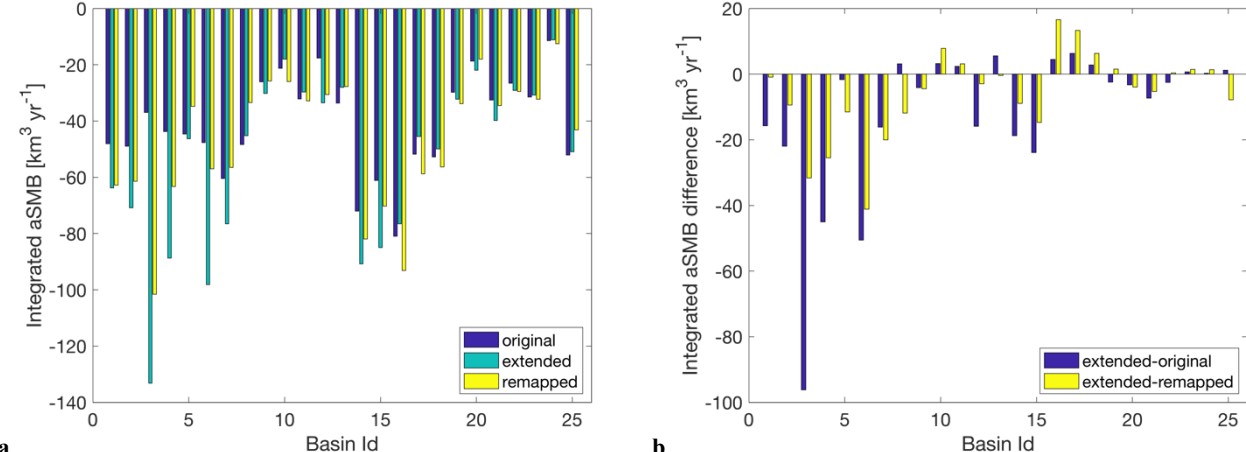

a                                           b

**Figure 8 Remapping results for a model state far from the observed geometry. (a) Integrated aSMB per basin from MAR model output on the observed ice mask (blue), for extension of the VUB_GISM model ice mask (green) and remapped to the VUB_GISM model geometry (yellow). (b) Differences between extended and original aSMB (blue) and between extended and remapped aSMB (yellow).**

## 4         Time dependent forcing

The same method can be used to define elevation-aSMB lookup tables and calculate remapped aSMB for climate change

15   scenarios, generating a time-dependent forcing. We have done this as a pilot application for MARv3.9 forced by MIROC5 (Watanabe et al. 2010) under scenario RCP8.5 (Figure 9) with available SMB data from 1950-2100 (Fettweis et al., 2013; Delhasse et al., 2019) computed for ISMIP6. We have calculated aSMB for the period 2015-2100 against a reference SMB as an average of the period 1960-1989. The resulting lookup tables (Figure 9) show the decrease in aSMB for the lower parts of each basin as expected.

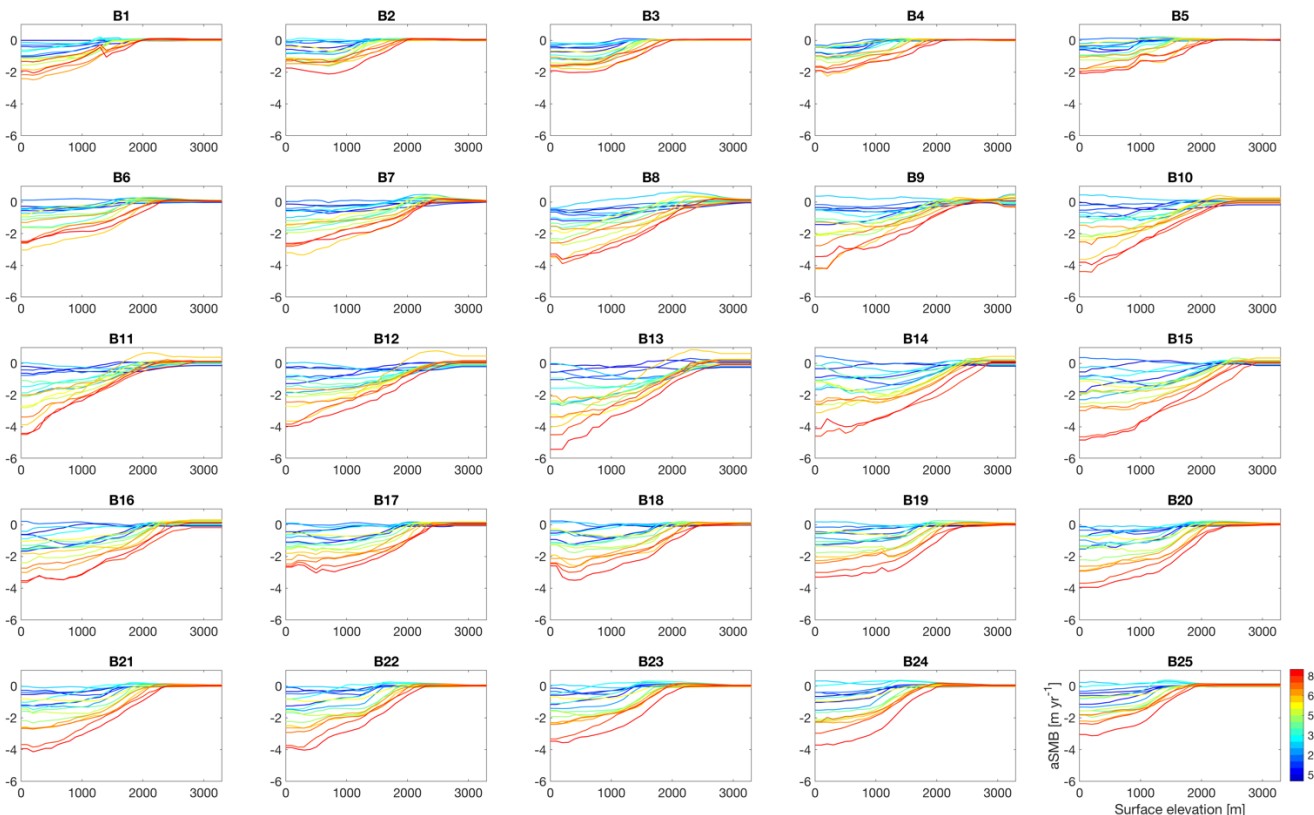

**Figure 9 Elevation-aSMB lookup tables for climate change scenario MAR MIROC5 RCP8.5. Time is colour coded to indicate years since 2015 with lines given every 5 years until year 2100. The subfigure titles indicate the basins as defined in Figure 2.**

## 4.1 Future sea-level change projections

The initial goal of the proposed method was to apply it to future sea-level change projections with a large ensemble of ice sheet models (with possibly widely different initial geometries) and forced by output of different climate models and scenarios, e.g. in the framework of the ice sheet model intercomparison project ISMIP6 (Nowicki et al., 2016; 2020; Goelzer et al., 2020). For such applications, the basin separation can be defined and the lookup tables can be calculated for specific climate models and scenarios ahead of time. Basin separation and weighting functions can be calculated for each specific ice sheet grid in advance. To apply a specific forcing scenario, the information transmitted to an individual ice sheet modeller consists of aSMB values for L elevation bands for M basins at N time steps. When the initial ice sheet geometries are known in advance, the remapping can also be done offline and aSMB(x,y,t) can be distributed directly, avoiding the need to implement the remapping in each individual ice sheet model (see section 2.2).

To test the feasibility of our method, we have applied it to a projection using only modelled and remapped aSMB to infer changes in ice sheet geometry. By ignoring any ice dynamic adjustment (i.e. no ice sheet model is used) and assuming the ice sheet to be in steady state with an unknown reference SMB, the time evolution of the ice sheet is fully determined by the initial geometry (surface elevation and mask) and the given aSMB. This setup does not consider any ice dynamic effects, such as the adjustment of ice flow to the SMB change itself and variations in marine terminating outlet glaciers. We emphasize that this experimental setup serves to illustrate the use of the remapping method and should not be interpreted as a full ice sheet projection including the dynamic response.

We first compare two different representations of the cumulative (time-integrated) SMB anomaly as a measure of the spatially resolved ice thickness change at the end of the scenario.

1. The time-integrated original aSMB of the climate model, by definition at fixed surface elevation (MOD).

2. The time-integrated aSMB calculated by remapping to the same fixed surface elevation (MAP).

In both cases, the resulting thickness change for aSMB<0 is limited by the available ice thickness at each grid point.

The two cases MOD and MAP show similar results (Figure 10a,b), indicating that the remapping performs well to capture the general pattern of SMB change also in this time-dependent application. Direct comparison between MOD and MAP (Figure 10c) reveal limitations in the remapping, mainly arising from localised melt and precipitation anomalies that are not resolved with 25 basins or where the relationship between surface elevation and aSMB breaks down (see also Figure 5c). The difference map (Figure 10c) shows some along-flow features on a larger spatial scale, suggesting that further refinement of the regions could improve the representation.

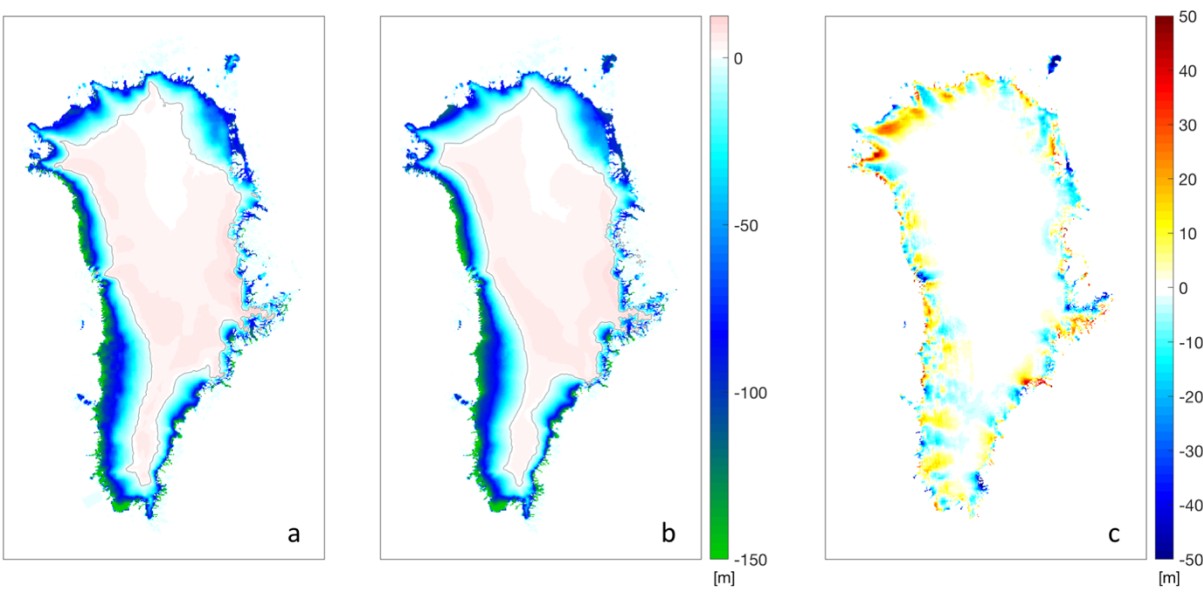

**Figure 10 Time-integrated aSMB for MOD (a), MAP (b) and differences MAP-MOD (c), representing the error of the remapping. The zero line is given in (a) and (b) as a grey contour.**

## 4.2 SMB-height feedback

In general the SMB anomaly that should be applied at any point on the evolving ice-sheet surface h depends both explicitly on

time t, because the climate is changing, and implicitly on time, because the ice-sheet surface h(t) is changing. The aim of this sub-section is to derive a method, including both effects, for estimating the SMB anomaly from RCM output, and to determine how this method can be applied in an ensemble of ice sheet models. In all other parts of this paper we have used "aSMB" for the SMB anomaly both in the RCM and as applied to the ice-sheet model. In this section (and Appendix A) alone, where the distinction is crucial, we reserve "SMB" and "aSMB" for quantities on the RCM grid, while by "ASMB" we mean the SMB

anomaly to be applied to the ice-sheet on its own surface h(t).

We denote the height by three symbols for different circumstances: $\bar{h}$ for the SMB anomaly and other quantities calculated from the RCM output at a fixed surface elevation, $h_0 = h(0)$ when remapping to the initial surface elevation that the ice-sheet has at $t = 0$, and $h = h(t)$ when remapping to a time evolving geometry. The SMB anomaly in the RCM (at fixed surface elevation $\bar{h}$) can then be expressed as $aSMB(t) = SMB(t) - SMB(0)$.

In order to perform the remapping, we first need to estimate a 3D field (including height-dependence) from the 2D field (at $\bar{h}$) given by the RCM. To do this, we need to estimate the local variation of SMB and aSMB with surface elevation i.e. $d(SMB(t))/dz$ and $d(aSMB(t))/dz$, respectively. The latter can be written as

$$d(aSMB(t))/dz = d(SMB(t))/dz - d(SMB(0))/dz, \qquad (6)$$

where the term $d(SMB)/dz(t)$ can be approximated from the RCM output, typically by analysing spatial SMB gradients in close proximity of the point of interest (Franco et al., 2012; Noël et al., 2016; Le clec'h et al., 2019), or by parameterising the

effect (e.g. Edwards et al., 2014a,b; Goelzer et al, 2013). Here, we derive $d(SMB)/dz(t)$ using MAR output (Franco et al., 2012).

The remapping of a time-dependent quantity X from the fixed RCM grid and fixed surface elevation $\bar{h}$ to some other ice-sheet surface Z may be formally written as an operator $R(X(t, \bar{h}), Z)$. Since the RCM surface $\bar{h}$, is fixed we will write the operator more simply as $R(X(t), Z)$ in the following. With this notation, the quantity used in the test procedure of Section 4.1 is

$R(aSMB(t), h_0)$, the time-evolving aSMB(t) remapped from the fixed RCM topography to the initial ice-sheet topography. This is *not* the SMB anomaly which should be applied to the time-evolving ice-sheet, because it includes only the climate-dependence of aSMB (its explicit dependence on time), and omits the effect of changing surface elevation (the implicit dependence on time via h(t)).

At first sight it may be surprising that the elevation effect is still *not* properly taken into account by the time-evolving aSMB(t)

remapped to the evolving h(t), $R(aSMB(t), h(t))$. This quantity involves a dependence on the modelled elevation change $dh(t) = h(t) - h_0$, and can be approximated as

$$R(aSMB(t), h) \approx R(aSMB(t), h_0) + R(d(aSMB(t))/dz, h_0) * dh(t). \tag{7}$$

By using (6), we get

$$R(aSMB(t), h) \approx R(aSMB(t), h_0) + [R(d(SMB(t))/dz, h_0) - R(d(SMB(0))/dz, h_0)] * dh(t) \tag{8}$$

(shown in **Figure 11**c). This quantity however includes only the elevation-dependence of the time-dependence of aSMB, which is a second-order effect, and it omits the first-order effect of the height feedback on SMB.

To preserve the full effect of elevation change on SMB, the quantity ASMB(h,t) that we need is the anomaly in remapped
5  SMB, rather than the remapped SMB anomaly $R(aSMB(t), h(t))$. The desired quantity is:

$$ASMB(t, h) \equiv R(SMB(t), h) - R(SMB(0), h_0) \tag{9}$$

$$\approx R(SMB(t), h_0) - R(SMB(0), h_0) + R(SMB(t), h) - R(SMB(t), h_0)$$

$$ASMB(t, h) \approx R(aSMB(t), h_0) + R(d(SMB(t))/dz, h_0) * dh(t). \tag{10}$$

Comparing (8) and (10), we can appreciate that (8) is incomplete because the first term in square brackets, which also appears in (10), is mostly cancelled by the second term in square brackets; indeed, if the vertical gradient of SMB is the same in the two climates, there is no effect of elevation change in (8).

To enable the calculation of (10) in ISMIP6, we remap the time-dependent $aSMB(t, \overline{h})$ and $d(SMB(t, \overline{h}))/dz$ to the initial ice-
10  sheet topography $h_0$. We have chosen this approach because the remapping can be done offline for a given initial ice sheet geometry. The format of data to be exchanged for an ensemble projection is then the same with and without remapping: the modeller receives time-dependent R(aSMB(x,y,t),h0) and R(d(SMB)/dz(x,y,t),h0) and has to implement a mechanism to calculate the additional term due to elevation change from the latter. An alternative online formulation, where the remapping would have to be implemented in each ice sheet model is given in Appendix A.

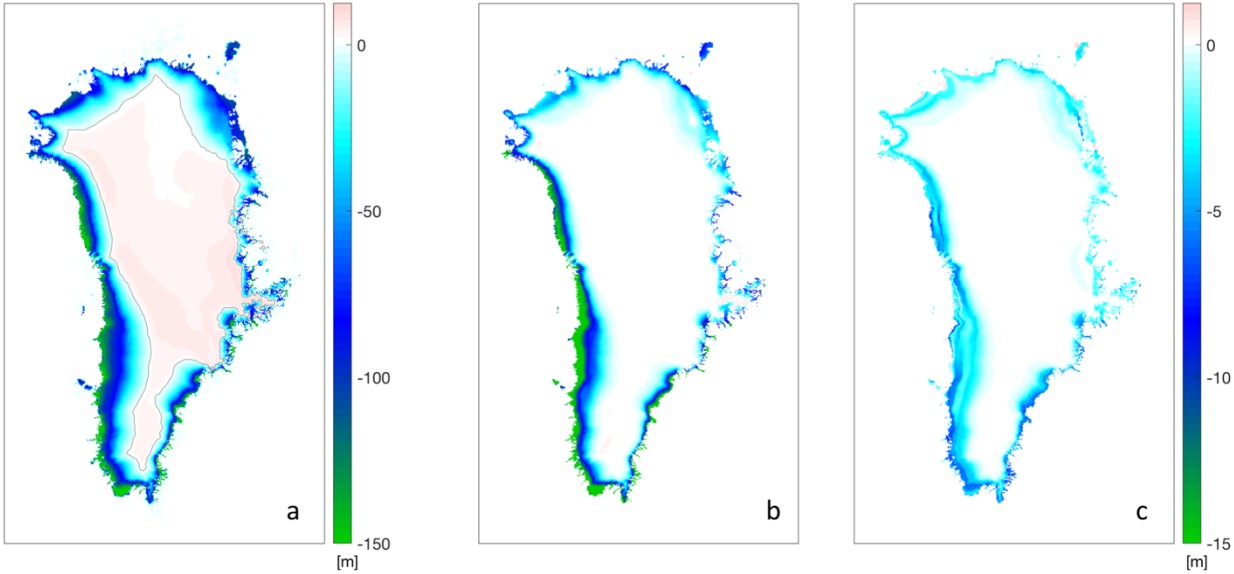

**Figure 11 Total elevation change 2015-2100 due to local time-integration of aSMB with remapping to the evolving geometry (a), elevation change due to d(SMB)/dz(t) (b) and due to remapping only (c). The zero line in a is given as grey contour. Note the different colour scale in (b) and (c) compared to (a).**

### 4.3 Application to a large ice sheet model ensemble

To illustrate the use of the proposed method (Eq. 10) for a larger group of models, we have applied the transient aSMB calculation for the modelled initial states of the initMIP-Greenland ensemble (Goelzer et al., 2018a). We use the publicly available output of the initial model states, which are provided on a common diagnostic grid (Goelzer et al., 2018b). The time-
10 dependent aSMB of MIROC5-forced MAR (RCP8.5) is remapped to the surface elevation of the initial state of each model. The geometry is then propagated (similar to section 4.1) over the period 2015-2100 as a function of the applied SMB anomaly (no ice sheet model is used), taking the height-SMB feedback into account as described in the last section. The resulting sea-level contribution (Figure 12a) is calculated by time-integration of the aSMB assuming an ocean surface area of $361.8 \times 10^6$ km$^2$ (Charette and Smith, 2010) and an ice density of 917 kg m$^{-3}$. Differences between models are due to differences in (initial)
15 ice sheet extent and surface elevation. We compare this result to a control experiment, with surface elevation changes considered as above, but here the original MAR aSMB is applied without remapping (Figure 12b).

Comparison between the two cases shows that (unphysical) biases in the estimated sea-level contribution are considerably reduced, especially for the models that show a too large initial ice sheet extent and consequently a too large sea-level contribution. However, some (physical) biases remain as expected, e.g. because a larger ice sheet has a larger ablation area.

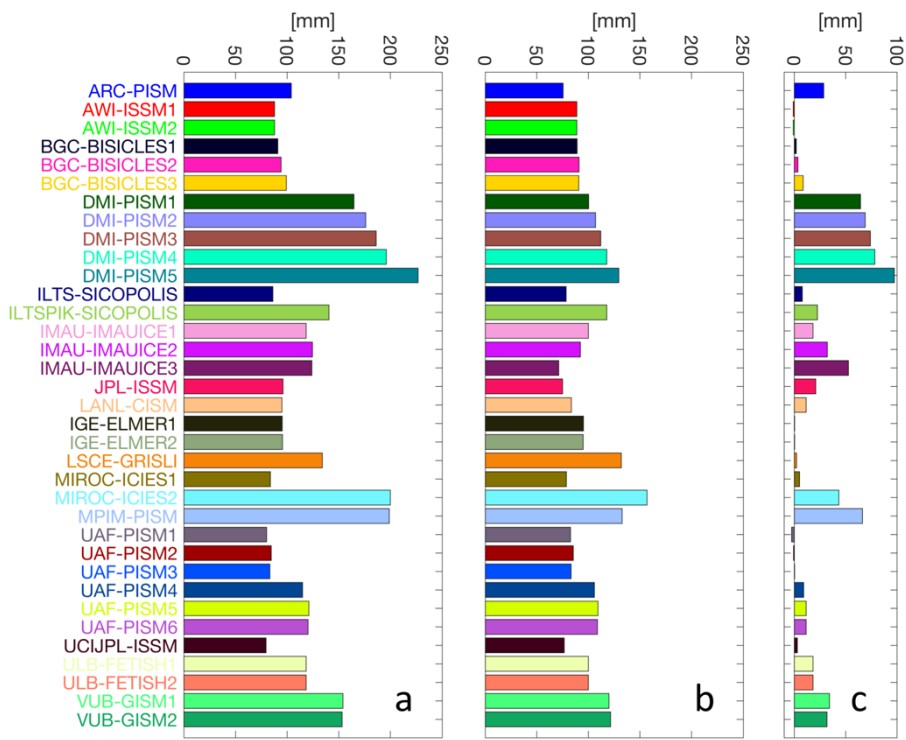

**Figure 12 Sea-level contribution in 2100 derived by integrating a transient aSMB over the initial ice mask of each initMIP-Greenland model, (a) without remapping, but extension to the modelled ice sheet extent, (b) with remapping to the initial surface elevation of each individual model and (c) difference (a)-(b).**

## 5   Discussion and conclusions

The described method allows application of SMB anomaly forcing for a large range of different ice sheet models and addresses problems arising from differences in initial ice sheet geometry. Remapping to the same geometry closely reproduces the original aSMB, while remapping to other, modelled geometries shows patterns similar to the original, with smooth and continuous aSMB across basin divides. This shows that the method is indeed suited to record and remap the aSMB for a wide range of ice sheet geometries, while retaining the physical patterns originally represented by the data.

Because the method produces a physically motivated aSMB forcing for a given ice sheet geometry, it also propagates biases in surface elevation to the SMB. This implies that for a given ice sheet geometry, biases due to a different ice sheet mask or due to elevation differences have to be accepted. In cases where the ice sheet mask is quite well matched, it may be preferred to apply aSMB without remapping to prevent propagation of small biases in surface elevation to the SMB forcing. In the initMIP-Greenland ensemble as a whole, biases due to differences in ice sheet mask were dominant, but this is not necessarily the case for each individual model. Therefore, we propose to evaluate the magnitude of the implied aSMB biases in offline calculations to decide whether remapping should be applied or not. This 'diagnostic mode' of the method can also be

envisioned for other applications, such as quantifying unphysical model biases for coupled and standalone ice sheet simulations.

The main difference between our method and existing approaches of transforming the SMB to a different geometry (Franco et al., 2012; Helsen et al., 2013) is the non-locality of the remapping process, which may be described as its key feature. Like Helsen et al., (2013) and Franco et al., (2012), we assume a linear relationship between elevation and SMB for a given time and location, but that relationship is not geographically uniform or constant in time. This means, however, that the original aSMB field is not exactly reproduced when the remapping is applied to an ice sheet with identical surface elevation, at least not for the basin delineation currently used. However, in the limit of reducing the width of the basins to individual flowlines, the reproduction of the aSMB at the original geometry should converge to the original field. Using a basin separation based on flow-lines is preferable, because they mostly follow the surface elevation gradient so the aSMB can be sampled in a continuous method that largely maintains the spatial structure. While this would increase the number of parameters that have to be fitted for each individual model geometry, it would also allow further improvement of the aSMB representation. We have based our delineation on an existing basin separation, but considerable handwork is required as long as automatic methods to generate meaningful basin separations of chosen detail for a complex geometry and flow like the GrIS are unavailable. We have tested the performance of the method for a schematic set of basins that can be more easily extended, albeit not following observed basin divides.

The ice sheet integrated mass anomaly is not conserved when remapping to a different geometry, given that a different geometry demands a different SMB forcing. It would in principle be possible to impose mass conservation on the ice sheet or even on the basin scale by comparing spatial averages of the original and remapped forcing and subtracting the difference. This would lead, however, to a spatial shift of regions where positive and negative anomalies are applied and, in the latter case, to discontinuities between neighbouring basins. Similar problems would arise for rescaling of aSMB.

We have shown how to apply the method for different ice sheet geometries, but so far have circumvented the problem of different model grids. While for ISMIP6 we have chosen to interpolate the already remapped aSMB to the native ice sheet model grids, the method could also be applied directly after interpolating the basin division and weighting to the individual ice sheet model grid. If the remapping were to be implemented in the ice sheet model itself, it could even be applied for adaptive grids that change over time.

On the input side, aSMB is provided in the present application at 5 km resolution, which was statistically downscaled from the regional climate model MAR run at 15 km. A similar grid resolution of the input data set should be envisioned when the aSMB comes instead from a coarse resolution GCM, because sufficient grid resolution is required to derive the lookup table for a chosen number of elevation bands. However, since remapping with a lookup table locally acts as a spatial linear

interpolator over the observed ice sheet, it propagates shortcomings of the input data set. The limiting factor for applying remapping to aSMB derived from GCMs or other coarse resolution models lies therefore in the quality of the original aSMB itself, rather than in technical aspects of the remapping.

The remapping is illustrated here with MAR v3.9 forced by MIROC5 as one of the data sets used in ISMIP6 projections (Goelzer et al., 2020). We have successfully applied the remapping also to output of the same MAR model forced by 5 other CMIP5 GCMs and 4 CMIP6 GCMs, and to output from an older MAR model version forced by 4 different GCMs. We therefore consider the remapping to be robust for a number of different forcing products.

## 6      Appendix A: Alternative formulation for the SMB-height feedback

An alternative method of calculating the dependence of ASMB on surface elevation (section 4.2) is described in the following. We can replace equations 9 and 10 by writing

$$ASMB(t, h) \equiv R(SMB(t), h) - R(SMB(0), h_0) \tag{11}$$

$$= R(SMB(t), h) - R(SMB(0), h) + R(SMB(0), h) - R(SMB(0), h_0)$$

$$ASMB(t, h) \approx R(aSMB(t), h) + R(d(SMB(0))/dz, h_0) * dh(t). \tag{12}$$

To calculate (12), we would have to remap the time-dependent $aSMB(t, \overline{h})$ and the initial $d(SMB(0))/dz$ to the time-evolving ice-sheet topography h. This implies that the remapping has to be implemented in the ice sheet model so that the lookup tables for both quantities can be applied online, in function of the changing geometry. From a practical point of view, the option

described in the main text (remap to a fixed initial elevation and apply $d(SMB)/dz(t)$, Eq.(10)) is much easier to achieve and has been chosen for the ISMIP6 projections (Nowicki et al., 2016; 2020; Goelzer et al., 2020). We have implemented and compared both methods in one ice sheet model and find nearly identical results for both of them.

*Code availability.* The scripts used for remapping, analysis and plotting are available at https://github.com/hgoelzer/aSMB-remapping and

will be saved in a publicly available archive on zenodo upon publication.

*Data availability.* The basin delineation and data sets used in this study are available during production at https://surfdrive.surf.nl/files/index.php/s/8nD5b2mMvG93sj1 and will be made publicly available on zenodo upon publication. The MAR based outputs for ISMIP6 are available at ftp://climato.be/fettweis/MARv3.9/ISMIP6. The initMIP ice sheet geometries are available at

https://doi.org/10.5281/zenodo.1173088.

*Author contribution.* HG conceived the study and developed the remapping method in discussion with the other authors. HG wrote the manuscript with assistance of the other authors.

*Competing interests.* Xavier Fettweis is a member of the editorial board of the journal.

*Acknowledgements*. We would like to thank Matthew Beckley for help with the extended basin delineation and Florian Ziemen for helpful discussion of early ideas for the proposed method. We acknowledge CMIP6 and the modelling groups participating in the initMIP-Greenland experiments of ISMIP6 for sharing their data and all members of the ISMIP6 team for discussions and feedback, with particular thanks to Sophie Nowicki and Tony Payne for their leadership.

Heiko Goelzer and Brice Noël have received funding from the program of the Netherlands Earth System Science Centre (NESSC), financially supported by the Dutch Ministry of Education, Culture and Science (OCW) under Grantnr. 024.002.001. Brice Noël acknowledges additional funding from the Netherlands Organisation for Polar Research (NWO). Computational resources for the MAR simulations performed for ISMIP6 have been provided by the Consortium des Équipements de Calcul Intensif (CÉCI), funded by the Fonds de la Recherche Scientifique de Belgique (F.R.S.–FNRS) under grant no. 2.5020.11 and the Tier-1 supercomputer (Zenobe) of the Fédération Wallonie Bruxelles infrastructure funded by the Walloon Region under the grant agreement no. 1117545. This material is based in part upon work supported by the National Center for Atmospheric Research, which is a major facility sponsored by the National Science Foundation under Cooperative Agreement No. 1852977.

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
