# Peer review of "Remapping of Greenland ice sheet surface mass balance anomalies for large ensemble sea-level change projections"

_The Cryosphere, 2019_

## Referee Comment (RC1) · Mario Krapp (Referee) · 7 Oct 2019

**Review**

This paper presents a method to correct for unphysical biases in the representation of the surface mass balance (SMB) in ice sheet model. By defining a remapping function for SMB anomalies for different drainage basins and height ranges, mismatches between ice sheet model geometry and climate model topography can be accounted for and thus leading to meaningful smooth and continuous SMB anomaly fields for diffferent geometries across basin divides. As a result of this approach the authors show that the SMB bias is reduced compared to commonly procedure of applying SMB anomalies. The paper focuses on standalone simulations of the GrIS with no interactive ice sheet.

The paper is well written and has a clear message how to address the SMB biases for different ice sheet models, specificically for the Greenland ice sheet. The paper targets a specific audience (ISMIP6 modellers) but as part of a special issues that is understandable.

The paper in its present form has a few shortcomings which need to be addressed before I can recommend it for publication. Find my general comments (and a couple of technical ones) below:

**General Comments**

- This approach works for relatively high-resolution climate model output, i.e., MAR SMB data (an RCM), but what about coarse-resolution GCMs? What would you regard as (spatial resolution) limit to the approach? E.g., the minimum required number of drainage basins depends on how many samples per basin/height bin your SMB resolution would provide.

- *Sensitivity on number of drainage basins*:
  What is the minimum number of drainage basins and/or height ranges that would give you an acceptable remapping? What is the overall sensititivy on resolution, i.e., number of basins? From figure 3, I can see that a at least a few different drainage basins could be grouped together, e.g., b3 & b4, b10-b13, b17 & 18, etc. I could see that fewer drainage basisn and fewer height ranges could give you similar I remapping error. To me, 25 drainage basins doesn't sound very "non-local" as is claimed in the abstract.

- *Uncertainty estimation*:
  The function aSMB=f(hc) isn't well constrained for a few drainage basins. b6, b13, or b25, for example, show substantial variations in each height bin. I don't

think they can be resolved if more basins would be used (also because the mapping becomes more local, which is not the intention) and also not with fewer, unfortunately. So, I if those variations can't be resolved they then needed to be accounted for at least. In b25, the aSMB variations around the median amount to more than 2m/yr. That is substantial. I would like to see how this inherent uncertainty in the remapping function plays out for the future projections. You say on page 8 that these uncertainties are small compared to the uncertainties in the climate forcing. Could you give some figures how large thos uncertainties in the climate model forcing are (in SMB units) and then compare it to you errors due to the remapping? Furthermore, if the remapping errors are relatively small, this could warrant using even fewer drainage basins (as I have suggested before).

- Could you also add the total integrated aSMB to Figure 6?

- On page 10 you say that the rempapping doesn't work well if the modelled surface elevation is too different from the observed. How "close" to observation does the ice modelled ice-sheet elevation need to be?

- ***Sect 4.1 (section naming)***
  If the setup "should not be interpreted as a real projection" (p.13 l.3), then you should revise the section header, as "**Future sea-level change projections**" implies realistic projections.

- ***p.13 ll.12***
  I don't think further refinement would help here. First of all, as you said, if it's a flowline feature than, of course it is a highly localised feature and depends on the representation of the ice sheet model. If you would increase the number of basins, your remapping becomes local, and you need to drop your claim about the key feature being its "non-locality" (see abstract)

- ***Sect 4.2***

I'm slightly confused by what is presented here and I must admit that it is not easy to follow. A little more motivation at the beginning of the paragraph could help to clarify what the actual problem is.

- – I thought that the overall goal of the remapping is to provide a transfer function from (any) h to aSMB. To me this implies that the changing ice geometry h(t) is implicity accounted for, or am I missing something?
- – Also, why is the reader presented with two methods, here?
- – Adding to my confusio, I don't understand what *Figure 11* tells me.
- – This section doesn't really "flow" with the rest of the manuscript and needs to be revised substantially
- – The authors favour the "oflline" version of the remapping. Wouldn't the interactive method make more sense as it would nudge the ice sheet towards realistic margins and elevation changes and thus reduce the mode bias on the go? I think that any model bias that is not corrected for (instantaneously) would lead to a model drift and thus ever-increasing errors. So, in my opinion, this method should be the recommended one.

- • *P15 l15* Which method is now the "proposed method"? Add the relevant reference to the Eq.

- • *Same page, ll19* "(no ice sheet model is used)" Does this imply (as bofore in Sect. 4.1) that the sea level contributions aren't based on realistic projections? Pleas, clarify.

- • *Sect 5 Discussion and conclusions*

  - – As far as I understand it, the remapping depends on the regional climate model used, i.e., MARv.9 forced by MIROC5. For the purpose of IS-MIP6, different GCM will be selected for the future projections (https://www.

the-cryosphere-discuss.net/tc-2019-191/). How sensible is the remapping approach to a different configuration of RCM forced by a GCM? While getting the correct figures is difficult, this needs to be discussed at least.

– How would the aSMB approach play out for "realistic" future projections.

– The paper is highly specific for a targeted audience, i.e., ISMIP6 modellers. I see that the remapping approach can reduce model biases and it is a quick method to run offline corrections after a standalon ice sheet model run. I can foresee that this method can also be used in a way as to quantify those "unphysical" model biases (by looking behind the reduction of the model bias we see in the integrated SMB responses of each ice sheet model in Figure 12), This is obviously beyond the scope of this paper but can be discussed.

• What can the reader expect for realistic large-ensemble sea-level change projections?

• I would highly recommend to make any scripts or tools that have been used for this paper publicly available.

• The available data are incomplete as are the data availability statements! Therefore, I couldn't review the associated data that have been produced along this paper. See the "data policy" section of TC (https://www.the-cryosphere.net/about/data_policy.html), for details

**Technical Comments**

• *P1 L27*: It is not clear what "observed" is referring to?

• I wouldn't use colors for the bar chart in *Figure 12*, as they don't add any information.

---

## Referee Comment (RC2) · Anonymous Referee #2 · 18 Oct 2019

General comments

This paper describes a method for adjusting a surface mass balance (SMB) anomaly field that has been calculated with reference to a specific ice sheet topography, such that it may be applied to an ice sheet model with a different topography, minimising unphysical impacts in the target model that may arise purely from this difference in initial height. In other words, it aims to estimate, from a single base field, the SMB anomaly that would be physically consistent with any given surface, without explicitly recalculating the climate and SMB anomalies on that surface. Such a method is desirable in a multi-model ice sheet comparison project such as ISMIP6, where a single future sce-

nario forcing needs to be applied consistently across a spectrum of ice sheet models that have significantly different representations of the present day ice sheet state.

Evidence from preliminary work in ISMIP (initMIP) shows that such a method will make comparing the ISMIP6 experiments across the different participating models very much more robust, so this is in principle a worthwhile contribution to the field, and may well prove useful beyond the immediate scope of the ISMIP6 experiments themselves. The method described is sensible and the paper is written carefully, and addresses the main questions that arise regarding its application and the degree to which it may inherently distort the input fields.

This is a methods paper, describing a method that I believe has already been used by a number of groups conducting the ISMIP6 experiments, so there's no real scientific interpretation to quibble over and the main goal of the paper is to document what was done, say why certain decisions were made how they were and enable others to reproduce the method. The only real fault I find with the current state of this paper then is the derivation of the time and height dependent remapping procedure, section 4.2. There's clearly some subtlety in how to remap the real changes in SMB that come from a changing climate (simulated in the RCM) at the same time as estimating what would be expected to physically occur as the ice sheet height evolves (not simulated in the RCM) along with applying the numerical dSMB/dz remapping to account for the initial state mismatch, but I found 4.2 a very confusing way of trying to explain this. This is perhaps due to the notation used - for me the summary in words at the end was much clearer than the form of the equations used to derive it. This section, alone, doesn't do a great job of allowing others to understand and reproduce the method for themselves.

Further, given that the method may have use beyond the current ISMIP6 effort, it might be useful to future readers to highlight things that could have been done better in hindsight, or that could be applied if a reader's individual use case isn't subject to some of the (wide-ranging) restrictions implied by the ISMIP ensemble. So, whilst it is stated (pg5, line 13) that other choices of hc and R might be appropriate for a non-

MAR forcing product, it might guide future applications for the authors to note some more detail as to why they decided their choice was "sufficient". In this vein, whilst there is attention on the needs of different ice sheet models as targets for the method, perhaps the authors could speculate on issues that might arise from using a different source climate model - eg a GCM with a lower horizontal resolution than MAR.

Lastly, playing devil's advocate (and supporting writing for maximum clarity, and defensively) readers not familiar with ISM modelling might question whether what's being done here is a fudge of the "right" boundary condition purely for the sake of convenience for modellers who haven't initialised their ice correctly and don't want it forced into correctness by these "right" boundary conditions. I would thus recommend being careful in outlining the motivation and the scientific intent of the adjustment. As an example, pg 2, l28: "appropriate" carries an ambiguous meaning here. I think there are a couple of other places terms like this are used too. For me it would be better to be very explicit and stress that the method is intended to transform the climate forcing so that it has more physical consistency with the ice sheet state it will be used with. So, in section 2 "this effect we are trying to capture" could be made more explicit along the lines of: here is a physical relationship between ice geometry and boundary conditions we need to be able to honour in each model that uses our forcing set, because the two things are not independent and blindly applying the same set of absolute boundary values to every ISM would impose an artificial inconsistency.

Detail comments

page 4, line1: "fixed function of observed surface elevation" could add "sampled across the entire ice sheet"

pg4, l5: "apply the remapping" could add "separately"

pg4, l13: it's not obvious to me why the median is used, rather than any other average

pg4, l16: is there any possibility at this point to recalculate/merge the drainage basins

for ISMs that might have very different/coarse geometries? Who does each part here? Do ISM groups remap themselves based on the lookup table?

pg4, l33: What were the MAR boundary conditions - ERA?

pg5, fig 2: The choice of colours is a bit random, some are indistinguishable from each other - does this invite unnecessary use of printer ink!?

pg5, l12: "judged sufficient" is not very precise - if you're going to say other intervals might be appropriate for other products, could you give some kind of guidance as to why you felt this choice was appropriate for this product?

pg9: might be a good place to note the effect of the remapping on (integrated) SMB conservation? One would not expect it to conserve, of course - and probably you actively don't want it to, again within a framework of transforming the SMB forcing so it gains physical consistency with the state you're applying it to rather than preserving numerical neatness for its own sake.

pg13, l12: notes "where the relationship between surface elevation and aSMB breaks down". I think this could ideally be expanded and come much earlier in the paper, as a general caveat to the applicability of the whole approach. Could you add an estimation of where/how badly this affects things, or how far the target topography can be from the original before this sort of method is not worth applying?

pg 14: whilst I got the principle fine, I found the derivation of the form of the time- and height-dependent anomaly on page 14 (ultimately, eqs (10) or (12)) to be very confusing. Not sure how best to suggest clarifying, but some points to consider: - Line 10, I didn't find the omission of $\hat{h}$ from the R(...) operator and elsewhere helpful for clarity, I ended up writing it back in everywhere it was not explicit to remind myself that these terms originated at $\hat{h}$ rather than any other h. For consistency throughout the uses of h, would $\hat{h}$ be more clear as $h\_RCM$, $h\_0$ as $h\_ISM(t=0)$, and $h(t)$ as $h\_ISM(t)$? - Lines 13-17 seem to be there primarily to illustrate what *not* to do, as a

misleading false start. Is this part really needed at all? - Line 18, and eqs (9) and (10) are the fairly straightforward aim of it all - could all of lines 10-17 actually be left out, and the two terms on the RHS of (10) just be explained as representing the explicit climate change dependence and the height-dependence of the SMB respectively (if that's what they are)? - The alternative form in eqs (11) and (12) is not uninteresting, but since it's ultimately not used I'm afraid it contributed more to my initial sense of confusion than my education. Could make it a footnote? - Line 6: I additionally wasn't clear how the various d(SMB)/dz terms were in practice derived for the ISMIP forcing product - via a local spatial SMB gradient from MAR, from the basin-scale SMB vs height lookup tables described in section 2.1 or one of the other methods noted in the references on lines 7 and 8?

pg16, l5: wasn't clear to me how the physical and unphysical biases in the sea-level contributions were being discriminated between, unless it's simply that the remapping is a good thing, so the biases left after applying it must be physical, and the difference between that and what was there before are unphysical?

pg17, l31: Are the scripts to do the remapping also going to be made available (with long-term storage) somewhere - perhaps as part of the TC submission?

---

## Referee Comment (RC3) · Anonymous Referee #3 · 22 Oct 2019

As part of ice sheet model intercomparison efforts, participating modeling groups utilize forcing fields such as anomalies of the surface mass balance (aSMB). These anomaly fields are constructed under the assumption that the ice sheet geometries (extent and surface height distribution) between the model and the reference are identical. If the geometries differ substantially, some remapping of the forcing fields is necessary to minimize unrealistic forcing. The authors present a compact and new procedure to remap atmospheric forcing fields, and they apply it to the Greenlandic ice sheet exemplarily.

The ice sheet is divided into sectors, which resemble here drainage basins of the ice

sheet. For each sector, they construct a lookup table of the actual aSMB and the (ice) surface elevation for defined elevation intervals from bottom to the top. The final lookup table contains the elevation-aSMB relationship for each basin. During the remapping, the applied relationships of the actual and neighboring basins are weighted according to their distance to the point of interest. The actual ice sheet elevation defines for each grid point the remapped forcing field. If this remapping is performed for all time steps, also transient aSMB fields could be remapped.

The authors show that the procedure works reasonably well for the trivial case, where the forcing field is remapped to the original reference topography. They also derive the influence of a temporarily evolving ice sheet elevation on the applied aSMB. Ultimately, they apply the procedure to model results of the initMIP exercise (Goelzer et al., 2018), where they analyze the formerly strongly diverging sea-level contributions of different models. These sea levels come closer together because the influence of the partly substantial different horizontal extent of the simulated ice sheets is corrected.

The manuscript is well-structured and written. I consider the reporting of scientific/technical procedures as important because they will help us to enhance the reproducibility of results and allows us to compare and understand diverging results. I recommend accepting this manuscript after minor revision.

**1   General comments**

The manuscript is well written and leaves only room for very few suggestions. The main assumption is the already mentioned strong dependence of the surface mass balance (SMB) with elevation. For the ablation part of the SMB, this is clear considering the strong relation between elevation and the near-surface air-temperature, where the latter could be understood as a proxy for melt potential. However, the same does not necessarily apply for the accumulation as part of the SMB. Could this difference disturb

your procedure? If yes, under which circumstances does it occur?

In some basins, you detect a substantial spread in the constructed primary lookup table (mid-east, south, north-west). Does a larger spread indicate that a further division of this section should be performed? Can you provide a criterion, that helps to weight the benefits of smaller basins and potentially smaller spread versus larger basins and larger spread?

You checked the sensitivity of $ds_{norm}$ for values between 50 km and 125 km and haven't found a strong dependence. What happens if $ds_{norm}$ reaches the grid resolution of the ice sheet model ($dxi_{sm}$): $ds_{norm} \longrightarrow dx_{ism}$? Do you detect beside discontinuities at the boundaries of the basins any other problem? What happens when the generally coarse grid of a driving global atmosphere model (resolution: $dx_{atm}$) is used, where we easily reach: $ds_{norm} \longrightarrow dx_{ism}$? This analysis may help to explain what happens if we drive the ice sheet directly with the output of global atmosphere models.

In the derivation of the SMB-height feedback, I have found the part (page 14) between lines 19 and 22 (incl.) confusing. May you move it into the supplement and refer to it for the interested reader, while the following "alternative" method becomes the main method.

The results of section 4.3 ("Application to a large ice sheet model ensemble") suggest that the correction (Figure 12c) is larger for models with a bigger initial sea-level contribution and ice-sheet extent. Have you tried to analyze the relation between $\frac{A - A_{ref}}{A_{ref}}$ and $\Delta z_{sl}$, where $A_{ref}$ and $A$ are the reference and actual ice-covered area in each model, respectively, and $\Delta z_{sl}$ is the sea-level difference (Figure 12c)? Please, at least add this figure to the supplements?

[Figure]

**2   Specific comments**

**2.1   Text**

**Page 3, Line 10**  What does "similar" actually mean? Please, clarify.

**Page 4, Line 13**  What happens if you use the mean instead of the median?

**Page 4, Line 23**  You may want to be more generic by replacing the "climate model's surface elevation" with the "reference field's elevation"?

**Page 6, Line 1**  I guess I understand you, but the sentence is not entirely clear. Please rephrase.

**Page 8, Line 8-9**  Here you state that the basins 7–9 have the largest mismatch. What is the reason behind?

**Page 9, Line 15**  Do you mean "where the modeled ice sheet is smaller (e.g. Basin 16, Figure 7d)"?

**Page 16, Line 1**  Please provide a citation for ocean area of $3.618 \cdot^{14}$ m$^2$?

**Page 17, Line 12**  The relationship is nearly uniform in each sector's center. You may clarify this if you think it's necessary.

**2.2   Figure**

The figures show in general the main features. However, some lines are hard to recognize, because they are too thin. Please check the figures.

**Figures 5, 7, 10, and 11:** In some of these figures, small deviations are hard to notice because the color around small deviations is white or gently yellow or light-blue. Would it be possible to use a color-bar, where either the deviation around zero is not white or, alternatively, mark the ocean with a light-gray color, for instance? If the ocean would be gray, you do not need to add a contour line to represent to coast.

**Figures 5 and 7:** The red contour lines are barely seen. Please thick the lines and mention its purpose in the related figure captions.

**Figure 3:** Please mention the meaning of the lower-right labels (basin number as defined in figure 2) in the figure caption.

**Figure 8, Subplot b:** It's hard to see if "extended-original" is as large as "remapped-extended?" Please replace "remapped-extended" with "extended-remapped."

**Figure 9:** Mention that each subfigure's title indicates the basins as defined in figure 2. The lower right subfigure is smaller due to the color-bar. Could you improve it? For example, by moving the color-bar to the right or below the group of subfigures.

**Figure 10, Subfigure a) and b):** Since the interior of Greenland shows only pale colors, you may add the zero contour line to guide the reader. If so, please mention the zero-contour line in the figure caption.

**Figure 11:** Since the interior of Greenland shows only pale colors, you may just add the zero contour line to guide the reader.

/sectionBibliography Goelzer, H., Nowicki, S., Edwards, T., Beckley, M., Abe-Ouchi, A., Aschwanden, A., Calov, R., Gagliardini, O., Gillet-Chaulet, F., Golledge, N. R., Gregory, J., Greve, R., Humbert, A., Huybrechts, P., Kennedy, J. H., Larour, E., Lipscomb, W.

H., Le clec'h, S., Lee, V., Morlighem, M., Pattyn, F., Payne, A. J., Rodehacke, C., Rückamp, M., Saito, F., Schlegel, N., Seroussi, H., Shepherd, A., Sun, S., van de Wal, R. and Ziemen, F. A.: Design and results of the ice sheet model initialisation initMIP-Greenland: an ISMIP6 intercomparison, Cryosph., 12(4), 1433–1460, doi:10.5194/tc-12-1433-2018, 2018.

---

## Author Comment (AC1) · 7 Mar 2020

We would like to thank the reviewers for their constructive comments that helped to improve the manuscript 'Remapping of Greenland ice sheet surface mass balance anomalies for large ensemble sea-level change projections'. We have revised the manuscript accordingly and would be happy to provide a new version.

Please find below the reviewer's comments in regular italic and a point-by-point response in bold font.

Referee 1 (Mario Krapp)

*This paper presents a method to correct for unphysical biases in the representation of the surface mass balance (SMB) in ice sheet model. By defining a remapping function for SMB anomalies for different drainage basins and height ranges, mismatches between ice sheet model geometry and climate model topography can be accounted for and thus leading to meaningful smooth and continuous SMB anomaly fields for diffferent geometries across basin divides. As a result of this approach the authors show that the SMB bias is reduced compared to commonly procedure of applying SMB anomalies. The paper focuses on standalone simulations of the GrIS with no interactive ice sheet.*
*The paper is well written and has a clear message how to address the SMB biases for different ice sheet models, specificically for the Greenland ice sheet. The paper targets a specific audience (ISMIP6 modellers) but as part of a special issues that is understandable.*
*The paper in its present form has a few shortcomings which need to be addressed before I can recommend it for publication. Find my general comments (and a couple of technical ones) below:*

**Thank you very much for the positive evaluation.**

*General Comments*
*• This approach works for relatively high-resolution climate model output, i.e., MAR SMB data (an RCM), but what about coarse-resolution GCMs? What would you regard as (spatial resolution) limit to the approach? E.g., the minimum required number of drainage basins depends on how many samples per basin/height bin your SMB resolution would provide.*

**The spatial resolution of the climate model is not a limiting factor for this method. If needed, the output of a GCM could easily be interpolated or downscaled to a higher grid resolution. For our application we have used a downscaled version of the MAR output, originally run at 15 km, subsequently downscaled to 1 km resolution and interpolated to 5 km for our analysis. We have added this information in the text**

and the advice that lower resolution model output should be interpolated to a higher resolution grid if needed. Whether the GCM provides a good enough representation of the SMB is a problem outside of the scope of our paper. For this it is important to remember that the remapping does not generate new information (except for regions outside the ice mask of the climate model). In terms of resolving the ice sheet SMB, the remapping can only be as good as the original SMB product. We have added a discussion point on that question in the manuscript

*• Sensitivity on number of drainage basins:*
*What is the minimum number of drainage basins and/or height ranges that would give you an acceptable remapping? What is the overall sensitivy on resolution, i.e., number of basins? From figure 3, I can see that a at least a few different drainage basins could be grouped together, e.g., b3 & b4, b10-b13, b17 & 18, etc. I could see that fewer drainage basisn and fewer height ranges could give you similar I remapping error. To me, 25 drainage basins doesn't sound very "non-local" as is claimed in the abstract.*

We would like to first clarify that our claim for the method to be 'non-local' refers to the remapping aspect of the procedure. What we mean is that the method can 'stretch' the original aSMB to the modelled geometry, as we have put it in the manuscript.

We have initially tested some examples of fewer than 25 basins, which led to unsatisfactory results, as the main climatological regions of the GrIS (dry SW/NE, wet SE/NW are not well captured. Notably, we have tested no separation (1 global lookup table) and 8 basins. In too large basins we typically find errors of opposite sign at opposite ends of the basin, suggesting that the height-aSMB relationship can be improved by further division. To formally explore sensitivity of the results to the number of drainage basins is difficult to achieve, because our initial delineation is handmade and a consistent variation of the number of basins is hard to do. Nevertheless, we have now added an evaluation of the number of basins in the supplement based on a schematic basin set that can more easily be extended. The limitation of this set is that it does not follow the observed basin divides, which is why we have maintained the original delineation in the main manuscript.

For the current delineation we have defined regions of roughly similar width around the ice sheet, as much as that was possible given the underlying drainage delineation (Mouginot et al. 2019). We believe re-combining basins for the sake of similarity would not make much sense, because it may limit robustness to other (untested) forcing fields and would only represent a very limited computational advantage.

*• Uncertainty estimation:*

*The function aSMB=f(hc) isn't well constrained for a few drainage basins. b6, b13, or b25, for example, show substantial variations in each height bin. I don't think they can be resolved if more basins would be used (also because the mapping becomes more local, which is not the intention) and also not with fewer, unfortunately. So, I if those variations can't be resolved they then needed to be accounted for at least. In b25, the aSMB variations around the median amount to more than 2m/yr. That is substantial. I would like to see how this inherent uncertainty in the remapping function plays out for the future projections. You say on page 8 that these uncertainties are small compared to the uncertainties in the climate forcing. Could you give some figures how large thos uncertainties in the climate model forcing are (in SMB units) and then compare it to you errors due to the remapping? Furthermore, if the remapping errors are relatively small, this could warrant using even fewer drainage basins (as I have suggested before).*

**As mentioned in response to the last question, we have included an analysis of the number of drainages basins in the supplement. The results show that some improvement can indeed be achieved with refining the basins further.**
**The aSMB error integrated over all basins is 19 km$^3$ yr$^{-1}$ or <1.7% for the end of the century SMB anomaly, as can now be read from figure 6 (see next point). This number compares to the CMIP5 ensemble range and standard deviation (6 models used in ISMIP6) of 650 km$^3$ yr$^{-1}$ and 240 km$^3$ yr$^{-1}$, respectively. This information has been added to the manuscript.**

*• Could you also add the total integrated aSMB to Figure 6?*

**OK, we have added the totals in the legend and updated the caption accordingly. The totals are also mentioned in the text in comparison to typical climate model uncertainty and it shows that the uncertainty is much smaller than the structural uncertainty underlying the aSMB calculation itself.**

*• On page10 you say that the rempapping doesn't work well if the modelled surface elevation is too different from the observed. How "close" to observation does the ice modelled ice-sheet elevation need to be?*

**There is no limit on how close the modelled elevation has to be to observations. The biases (in the interior) are simply proportional to the height differences and the**

**SMB gradient in the forcing model. The point here is to clarify that the "feature of the remapping method […] can be interpreted both as an asset or as a shortcoming": the forcing is 'corrected' to the modelled surface elevation. We pick up on that point in the discussion.**

• *Sect 4.1 (section naming)*
*If the setup "should not be interpreted as a real projection" (p.13 l.3), then you should revise the section header, as "Future sea-level change projections" implies realistic projections.*

**OK, replaced 'real projection' by 'full ice sheet projection' at p13 L3. We just want to avoid that the SL numbers are taken as ice sheet contributions in global assessments, because we only integrate SMB anomalies and do not consider any ice sheet dynamics.**

• *p.13 ll.12*
*I don't think further refinement would help here. First of all, as you said, if it's a flowline feature than, of course it is a highly localised feature and depends on the representation of the ice sheet model. If you would increase the number of basins, your remapping becomes local, and you need to drop your claim about the key feature being its "non-locality" (see abstract)*

**There is a misunderstanding with the term 'non-local' here. See also comment above. What we assert with 'non-local' is the feature of the reconstruction part of the method to translate the SMB anomaly found for one place to another. This applies equally for a flowline that would be 'stretched' (or compressed) to fit the modelled topography. We maintain that the refinement would help to better resolve aSMB variations perpendicular to the flow-direction, most obvious for the basins 9, 15, 16 ,17, but also 2 and 3 (Fig. 5c). This discussion has been added in the manuscript.**

• *Sect 4.2*
*I'm slightly confused by what is presented here and I must admit that it is not easy to follow. A little more motivation at the beginning of the paragraph could help to clarify what the actual problem is.*

**We know from our own experience and discussion that the problem is involved and (we believe inherently) difficult to understand. We further clarified the description e.g. by adding a statement what to expect from this sub-section up front as suggested.**

*– I thought that the overall goal of the remapping is to provide a transfer function from (any) h to aSMB. To me this implies that the changing ice geometry h(t) is implicity accounted for, or am I missing something?*

**At the beginning of this work, we had the same understanding, but realised that this was not the case. The changing ice geometry h(t) is implicitly accounted for in its effect on *aSMB*, but not on the SMB itself. To clarify this distinction and include changes in SMB due to height changes is the point of this sub-section.**

*– Also, why is the reader presented with two methods, here?*

**We agree that presenting both methods was confusing, so we have moved the description of the second method to the appendix, following the advice of all three referees.**

*– Adding to my confusio, I don't understand what Figure 11 tells me.*

**The figure simply shows the additional effect we are including when adding the height-SMB feedback in addition to adding the dependence of aSMB on h(t).**

*– This section doesn't really "flow" with the rest of the manuscript and needs to be revised substantially*

**We have revised the text to make it more clear and accessible.**

*– The authors favour the "offline" version of the remapping. Wouldn't the interactive method make more sense as it would nudge the ice sheet towards realistic margins and elevation changes and thus reduce the mode bias on the go? I think that any model bias that is not corrected for (instantaneously) would lead to a model drift and thus ever-increasing errors. So, in my opinion, this method should be the recommended one.*

**The two methods are in theory identical, we have implemented both in one of the ice sheet models and find near identical results. As described, we favour the offline version for practical purposes.**

**Nudging an ice sheet model towards realistic margins would create a model drift during the projections, which is clearly not intended, so there may be a general misunderstanding about the role of the aSMB forcing.**

*• P15 l15 Which method is now the "proposed method"? Add the relevant reference to the Eq.*

**The whole section has been revised and the alternative method has been moved to the appendix. We have included reference to equation 10 as the proposed method.**

*• Same page, ll19 "(no ice sheet model is used)" Does this imply (as bofore in Sect. 4.1) that the sea level contributions aren't based on realistic projections? Pleas, clarify.*

**The SMB projections are real and represent a possible forcing for ice sheet model experiments. See response to comments (Sect. 4.1) before.**

*• Sect 5 Discussion and conclusions*
*– As far as I understand it, the remapping depends on the regional climate model used, i.e., MARv.9 forced by MIROC5. For the purpose of ISMIP6, different GCM will be selected for the future projections (https://www.the-cryosphere-discuss.net/tc-2019-191/). How sensible is the remapping approach to a different configuration of RCM forced by a GCM? While getting the correct figures is difficult, this needs to be discussed at least.*

**We have initially tested the remapping with an earlier MAR version forced by 4 different GCMs, and by now also successfully used the method with MARv.9 forced by 9 different CMIP GCMs. The basin delineation holds for these other forcing fields as well. We have added this information as a discussion point.**

*– How would the aSMB approach play out for "realistic" future projections.*

**The example we show is a realistic projection in terms of SMB. The reason we emphasize that it is not a sea-level projection is that we did not run an ice sheet model, so ice dynamics are not included. Once an aSMB forcing is remapped to the**

**specific ice sheet geometry, it produces results like any other aSMB forcing. By now, the remapping has also been successfully applied for the ISMIP6 projections.**

*– The paper is highly specific for a targeted audience, i.e., ISMIP6 modellers. I see that the remapping approach can reduce model biases and it is a quick method to run offline corrections after a standalon ice sheet model run. I can foresee that this method can also be used in a way as to quantify those "unphysical" model biases (by looking behind the reduction of the model bias we see in the integrated SMB responses of each ice sheet model in Figure 12), This is obviously beyond the scope of this paper but can be discussed.*

**Yes, agreed. The remapping may be used diagnostically and also in any other case where the ice sheet model geometry differs from the geometry of the climate model. We have added a discussion point along those lines.**

*• What can the reader expect for realistic large-ensemble sea-level change projections?*

**The reader can expect the addition of ocean forcing, ice dynamics responding to the forcing and a wide range of ice sheet models. The projection paper using the method has appeared in TC discussions. See also points before.**

*• I would highly recommend to make any scripts or tools that have been used for this paper publicly available.*

**Yes, we fully agree. The scripts are already available on github and will be archived as a zenodo archive upon publication.**

*• The available data are incomplete as are the data availability statements! Therefore, I couldn't review the associated data that have been produced along this paper. See the "data policy" section of TC (https://www.the-cryosphere.net/about/ data_policy.html), for details*

**Scripts and datasets are now available online and will be archived for publication. We have modified the availability statements accordingly.**

*Technical Comments*
*• P1 L27: It is not clear what "observed" is referring to?*

**OK, replaced 'observed geometry' by 'climate model geometry'. In our framework and aside from climate model resolution they are identical.**

*• I wouldn't use colors for the bar chart in Figure 12, as they don't add any information.*

**The colour scheme has been chosen to be in line with figures in the initMIP paper and facilitate direct comparison. Not changed.**

**Thanks again for reviewing this paper.**

*Referee 2*

*General comments*

*This paper describes a method for adjusting a surface mass balance (SMB) anomaly field that has been calculated with reference to a specific ice sheet topography, such that it may be applied to an ice sheet model with a different topography, minimising un-physical impacts in the target model that may arise purely from this difference in initial height. In other words, it aims to estimate, from a single base field, the SMB anomaly that would be physically consistent with any given surface, without explicitly recalculating the climate and SMB anomalies on that surface. Such a method is desirable in a multi-model ice sheet comparison project such as ISMIP6, where a single future scenario forcing needs to be applied consistently across a spectrum of ice sheet models that have significantly different representations of the present day ice sheet state.*
*Evidence from preliminary work in ISMIP (initMIP) shows that such a method will make comparing the ISMIP6 experiments across the different participating models very much more robust, so this is in principle a worthwhile contribution to the field, and may well prove useful beyond the immediate scope of the ISMIP6 experiments themselves. The method described is sensible and the paper is written carefully, and addresses the main questions that arise regarding its application and the degree to which it may inherently distort the input fields.*

**Thank you very much for the positive evaluation.**

*This is a methods paper, describing a method that I believe has already been used by a number of groups conducting the ISMIP6 experiments, so there's no real scientific*

*interpretation to quibble over and the main goal of the paper is to document what was done, say why certain decisions were made how they were and enable others to reproduce the method. The only real fault I find with the current state of this paper then is the derivation of the time and height dependent remapping procedure, section 4.2. There's clearly some subtlety in how to remap the real changes in SMB that come from a changing climate (simulated in the RCM) at the same time as estimating what would be expected to physically occur as the ice sheet height evolves (not simulated in the RCM) along with applying the numerical dSMB/dz remapping to account for the initial state mismatch, but I found 4.2 a very confusing way of trying to explain this. This is perhaps due to the notation used - for me the summary in words at the end was much clearer than the form of the equations used to derive it. This section, alone, doesn't do a great job of allowing others to understand and reproduce the method for themselves.*

**We have reworked section 4.2 by starting with a clear motivation and by moving the alternative formulation to the appendix. See also response to several comments raised by reviewer 1.**

*Further, given that the method may have use beyond the current ISMIP6 effort, it might be useful to future readers to highlight things that could have been done better in hindsight, or that could be applied if a reader's individual use case isn't subject to some of the (wide-ranging) restrictions implied by the ISMIP ensemble. So, whilst it is stated (pg5, line 13) that other choices of hc and R might be appropriate for a non-MAR forcing product, it might guide future applications for the authors to note some more detail as to why they decided their choice was "sufficient". In this vein, whilst there is attention on the needs of different ice sheet models as targets for the method, perhaps the authors could speculate on issues that might arise from using a different source climate model - eg a GCM with a lower horizontal resolution than MAR.*

**We have clarified in the text the main factors influencing the parameter choices and why we have considered them to be 'sufficient'. We have also added a discussion item to address other use cases, e.g. when applying a different climate model. See also response to comment from Reviewer 1.**

*Lastly, playing devil's advocate (and supporting writing for maximum clarity, and defensively) readers not familiar with ISM modelling might question whether what's being done here is a fudge of the "right" boundary condition purely for the sake of convenience for modellers who haven't initialised their ice correctly and don't want it*

*forced into correctness by these "right" boundary conditions. I would thus recommend being careful in outlining the motivation and the scientific intent of the adjustment. As an example, pg 2, l28: "appropriate" carries an ambiguous meaning here. I think there are a couple of other places terms like this are used too. For me it would be better to be very explicit and stress that the method is intended to transform the climate forcing so that it has more physical consistency with the ice sheet state it will be used with. So, in section 2 "this effect we are trying to capture" could be made more explicit along the lines of: here is a physical relationship between ice geometry and boundary conditions we need to be able to honour in each model that uses our forcing set, because the two things are not independent and blindly applying the same set of absolute boundary values to every ISM would impose an artificial inconsistency.*

**Good point. We have revisited the text to work it towards more careful formulations. We have replaced 'appropriate' by 'physically consistent' on page 2 and revised the description in section 2 as suggested.**

*Detail comments*
*page 4, line1: "fixed function of observed surface elevation" could add "sampled across the entire ice sheet"*

**Thanks, added as suggested.**

*pg4, l5: "apply the remapping" could add "separately"*

**OK, added.**

*pg4, l13: it's not obvious to me why the median is used, rather than any other average*

**We tried the average and found the median to be more robust against eventual outliers. This has been added in the text.**

*pg4, l16: is there any possibility at this point to recalculate/merge the drainage basins for ISMs that might have very different/coarse geometries? Who does each part here? Do ISM groups remap themselves based on the lookup table?*

**The drainage basin delineation should not change between analysis and reconstruction to avoid distortion of the aSMB field. We at least do not see a**

**meaningful way to merge different basins. The remapping could in principle be done by each individual modeller. In practice, we have provided remapped SMB based on individual initial ice sheet geometries. This is described in more detail in section 4.2.**

*pg4, l33: What were the MAR boundary conditions - ERA?*

**MAR was forced by MIROC5. We have added 'forced by MIROC5 (Watanabe et al. 2010)'.**

*pg5, fig 2: The choice of colours is a bit random, some are indistinguishable from each other - does this invite unnecessary use of printer ink!?*

**OK, we have updated the figure as a grey scale image instead.**

*pg5, l12: "judged sufficient" is not very precise - if you're going to say other intervals might be appropriate for other products, could you give some kind of guidance as to why you felt this choice was appropriate for this product?*

**OK. We have expanded on this point to clarify why we judge the parameter choice as sufficient. This is related to the spatial variability and smoothness of the original aSMB product.**

*pg9: might be a good place to note the effect of the remapping on (integrated) SMB conservation? One would not expect it to conserve, of course - and probably you actively don't want it to, again within a framework of transforming the SMB forcing so it gains physical consistency with the state you're applying it to rather than preserving numerical neatness for its own sake.*

**OK, added a sentence clarifying this relation:**
**'This also illustrates why the method is not designed to conserve mass when remapping to a different geometry: it demands a different SMB forcing.'**

*pg13, l12: notes "where the relationship between surface elevation and aSMB breaks down". I think this could ideally be expanded and come much earlier in the paper, as a general caveat to the applicability of the whole approach.*

**Agreed. We have now added comments about this limitations already on page 9 where results presented in Figure 5 are discussed.**

*Could you add an estimation of where/how badly this affects things, or how far the target topography can be from the original before this sort of method is not worth applying?*

**The discussion in this point deals with errors when remapping to the same geometry, not to a different geometry. We have an extended discussion point on the limitations for different geometries. See also comments to reviewer 1.**

*pg 14: whilst I got the principle fine, I found the derivation of the form of the time- and height-dependent anomaly on page 14 (ultimately, eqs (10) or (12)) to be very confusing. Not sure how best to suggest clarifying, but some points to consider:*

**This sub-section was also criticised by the other reviewers and we have reworked it accordingly. With some comments overlapping, please also see responses to the other reviewers.**

*Line 10, I didn't find the omission of hˆbar from the R(...) operator and elsewhere helpful for clarity, I ended up writing it back in everywhere it was not explicit to remind myself that these terms originated at hˆbar rather than any other h. For consistency throughout the uses of h, would hˆbar be more clear as h_RCM, h_0 as h_ISM(t=0), and h(t) as h_ISM(t)?*

**We feel the h^bar in R(…) is redundant, because any remapping operation is always from h^bar to another h. There is a balance between adding more information to the symbols and keeping the formulation compact, the latter of which we have emphasized by choosing single letter symbols where possible. We are confident that other changes to this part have made the sub-section much clearer. Though, we've learned by own experience that this part of the remapping is very difficult to understand.**

*Lines 13-17 seem to be there primarily to illustrate what \*not\* to do, as a misleading false start. Is this part really needed at all?*
*Line 18, and eqs (9) and (10) are the fairly straightforward aim of it all - could all of lines 10-17 actually be left out, and the two terms on the RHS of (10) just be explained as*

*representing the explicit climate change dependence and the height-dependence of the SMB respectively (if that's what they are)?*

**We feel it is important to lay out the whole complexity of the problem to make readers understand this is a non-trivial issue. We have tried to improve clarity by revising the description.**

*The alternative form in eqs (11) and (12) is not uninteresting, but since it's ultimately not used I'm afraid it contributed more to my initial sense of confusion than my education. Could make it a footnote?*

**Agreed. We have moved this part to the Appendix for interested readers.**

*Line 6: I additionally wasn't clear how the various d(SMB)/dz terms were in practice derived for the ISMIP forcing product - via a local spatial SMB gradient from MAR, from the basin-scale SMB vs height lookup tables described in section 2.1 or one of the other methods noted in the references on lines 7 and 8?*

**We use the Franco method based on MAR output. This has been clarified in the text.**

*pg16, l5: wasn't clear to me how the physical and unphysical biases in the sea-level contributions were being discriminated between, unless it's simply that the remapping is a good thing, so the biases left after applying it must be physical, and the difference between that and what was there before are unphysical?*

**The physical and unphysical biases can indeed not formally be discriminated. Though the idea of the argument is to contrast biases that can be (physically) expected for an ice sheet of different shape and the (unphysical) biases of an aSMB derived for one geometry directly applied to another.**

*pg17, l31: Are the scripts to do the remapping also going to be made available (with long-term storage) somewhere - perhaps as part of the TC submission?*

**Yes, they will be made publicly available through links in the availability section.**

**Thanks again for a constructive review of our manuscript.**

*Referee 3*

*As part of ice sheet model intercomparison efforts, participating modeling groups utilize forcing fields such as anomalies of the surface mass balance (aSMB). These anomaly fields are constructed under the assumption that the ice sheet geometries (extent and surface height distribution) between the model and the reference are identical. If the geometries differ substantially, some remapping of the forcing fields is necessary to minimize unrealistic forcing. The authors present a compact and new procedure to remap atmospheric forcing fields, and they apply it to the Greenlandic ice sheet exemplarily.*

*The ice sheet is divided into sectors, which resemble here drainage basins of the ice sheet. For each sector, they construct a lookup table of the actual aSMB and the (ice) surface elevation for defined elevation intervals from bottom to the top. The final lookup table contains the elevation-aSMB relationship for each basin. During the remapping, the applied relationships of the actual and neighboring basins are weighted according to their distance to the point of interest. The actual ice sheet elevation defines for each grid point the remapped forcing field. If this remapping is performed for all time steps, also transient aSMB fields could be remapped.*

*The authors show that the procedure works reasonably well for the trivial case, where the forcing field is remapped to the original reference topography. They also derive the influence of a temporarily evolving ice sheet elevation on the applied aSMB. Ultimately, they apply the procedure to model results of the initMIP exercise (Goelzer et al., 2018), where they analyze the formerly strongly diverging sea-level contributions of different models. These sea levels come closer together because the influence of the partly substantial different horizontal extent of the simulated ice sheets is corrected.*

*The manuscript is well-structured and written. I consider the reporting of scientific/technical procedures as important because they will help us to enhance the re-producibility of results and allows us to compare and understand diverging results. I recommend accepting this manuscript after minor revision.*

**Thanks for the positive comments.**

*1 General comments*

*The manuscript is well written and leaves only room for very few suggestions. The main assumption is the already mentioned strong dependence of the surface mass balance (SMB) with elevation. For the ablation part of the SMB, this is clear considering the strong relation between elevation and the near-surface air-temperature, where the latter could be understood as a proxy for melt potential. However, the same does not*

*necessarily apply for the accumulation as part of the SMB. Could this difference disturb your procedure? If yes, under which circumstances does it occur?*

**The remapping is most important for the margins of the ice sheet, where ablation is typically the dominant term to the SMB and aSMB. In the accumulation area in the interior of the GrIS, the elevation dependence of aSMB decreases, but aSMB is anyway close to zero, which mitigates this effect.**
**Note also that we operate with a lookup table that is *mapped* as a function of elevation, not with a regression. This implies that the method can also deal with the situation where aSMB decreases with decreasing elevation and then increases again (e.g. basin 2 in Fig. 3), which is not the case e.g. in the method of Helsen et al. (2013).**

*In some basins, you detect a substantial spread in the constructed primary lookup table (mid-east, south, north-west). Does a larger spread indicate that a further division of this section should be performed? Can you provide a criterion, that helps to weight the benefits of smaller basins and potentially smaller spread versus larger basins and larger spread?*

**Indeed, further refinement of the basins can improve the representation. We have included an analysis of the number of basins in the supplement with an alternative schematic basin delineation. The main problem to move to a larger number of basins is the difficulty to define a meaningful basin set. This is now discussed in the manuscript. See also responses to reviewer 1.**

*You checked the sensitivity of dsnorm for values between 50 km and 125 km and haven't found a strong dependence. What happens if dsnorm reaches the grid resolution of the ice sheet model (dxism): dsnorm −→ dxism?*

**The grid resolution of the aSMB product in use is 5 km, quite far outside of the interesting range for dsnorm. Nevertheless, as pointed out in response to questions by reviewer 1, the aSMB product could simply be interpolated or downscaled to a higher resolution to avoid technical issues if *dsnorm −→ dxism*. See also response to reviewer 1 on this question.**

*Do you detect beside discontinuities at the boundaries of the basins any other problem?*

**For some basins the tables would not have entries at high elevation due to limited coverage of the elevation range. This is already mentioned in the text.**

*What happens when the generally coarse grid of a driving global atmosphere model (resolution: dxatm) is used, where we easily reach: dsnorm $\longrightarrow$ dxism?*
*This analysis may help to explain what happens if we drive the ice sheet directly with the output of global atmosphere models.*

**The grid resolution itself is not a limiting factor, as the data can be interpolated to sufficient resolution to fulfil technical requirements for the remapping. Instead, the real limitation is in the quality of the original aSMB product. We have added a discussion item on that question.**

*In the derivation of the SMB-height feedback, I have found the part (page 14) between lines 19 and 22 (incl.) confusing. May you move it into the supplement and refer to it for the interested reader, while the following "alternative" method becomes the main method.*

**Line 19-22 describes the preferred method that has actually been used, so we want to keep that in the main text. We have moved the alternative method (line 23-) to the appendix to make the section clearer.**

*The results of section 4.3 ("Application to a large ice sheet model ensemble") suggest that the correction (Figure 12c) is larger for models with a bigger initial sea-level contribution and ice-sheet extent. Have you tried to analyze the relation between $(A-Aref)/Aref$ and $\Delta zsl$, where Aref and A are the reference and actual ice-covered area in each model, respectively, and $\Delta zsl$ is the sea-level difference (Figure 12c)? Please, at least add this figure to the supplements?*

**Indeed, there is a clear tendency for models with a larger area to exhibit larger corrections in Figure 12c. A scatter plot of this relationship is given below. However, while this tendency is mentioned in the manuscript, we found the figure does not add important information and have not included it.**

[Figure]

**Figure 1 Initial ice sheet area of initMIP-Greenland models (Goelzer et al., 2018) and difference of sea-level contribution when remapping is applied (Fig 12c).**

*2 Specific comments*
*2.1 Text*

*Page 3, Line 10 What does "similar" actually mean? Please, clarify.*

**OK, Replaced 'similar' by 'close'**

*Page 4, Line 13 What happens if you use the mean instead of the median?*

**The median is chosen because it is more robust against outliers. This has been added in the text.**

*Page 4, Line 23 You may want to be more generic by replacing the "climate model's surface elevation" with the "reference field's elevation"?*

**OK, replaced as suggested.**

*Page 6, Line 1 I guess I understand you, but the sentence is not entirely clear. Please rephrase.*

**OK, reformulated.**

*Page 8, Line 8-9 Here you state that the basins 7–9 have the largest mismatch. What is the reason behind?*

**We have added following description to the text: 'These three basins all exhibit detailed and varied topography at the margins, which may contribute to the errors. The largest signed errors are found in basin 7 with compensating biases of opposite sign.'**

*Page 9, Line 15 Do you mean "where the modeled ice sheet is smaller (e.g. Basin 16, Figure 7d)"?*

**Yes, thanks for the suggestion.**

*Page 16, Line 1 Please provide a citation for ocean area of 3.618 ·14 m2?*

**OK, added a reference.**

*Page 17, Line 12 The relationship is nearly uniform in each sector's center. You may clarify this if you think it's necessary.*

**Not necessary as it does not contradict the linearity mentioned here.**

*2.2 Figure*
*The figures show in general the main features. However, some lines are hard to recognize, because they are too thin. Please check the figures.*

**OK, we have checked all figures and redrawn and removed in some cases the contour lines. See details below.**

*Figures 5, 7, 10, and 11: In some of these figures, small deviations are hard to notice because the color around small deviations is white or gently yellow or light-blue. Would it be possible to use a colorbar, where either the deviation around zero is not white or, alternatively, mark the ocean with a light-gray color, for instance? If the ocean would be gray, you do not need to add a contour line to represent to coast.*

**OK, we have marked the ocean in gray colour and kept the present-day coastline only in some cases where that information seems useful.**

*Figures 5 and 7: The red contour lines are barely seen. Please thick the lines and mention its purpose in the related figure captions.*

**OK, we have removed the contour lines in Fig 5, where they did not add more information. For Fig 7 we have made the lines thicker and grey to better match with the background colour and avoid a colour that is in one of the colourbars.**

*Figure 3: Please mention the meaning of the lower-right labels (basin number as defined in figure 2) in the figure caption.*

**OK. added description in the caption.**

*Figure 8, Subplot b: It's hard to see if "extended-original" is as large as "remapped-extended?" Please replace "remapped-extended" with "extended-remapped."*

**OK. adapted as suggested.**

*Figure 9: Mention that each subfigure's title indicates the basins as defined in figure 2.*

**OK, this information is added in the caption.**

*The lower right subfigure is smaller due to the color-bar. Could you improve it? For example, by moving the color-bar to the right or below the group of sub- figures.*

**OK, we have updated the figure accordingly.**

*Figure 10, Subfigure a) and b): Since the interior of Greenland shows only pale colors, you may add the zero contour line to guide the reader. If so, please mention the zero-contour line in the figure caption.*

**OK, updated as suggested.**

*Figure 11: Since the interior of Greenland shows only pale colors, you may just add the zero contour line to guide the reader.*

OK, we have added a contour line in panel a. We have tried the same for panel b and c but found it decreased readability of the figures. So we keep it only in panel a.

Thank you very much for the review.